# N-cadherin-regulated FGFR ubiquitination and degradation control mammalian neocortical projection neuron migration

Elif Kon[1], Elisa Calvo-Jiménez[1], Alexia Cossard[1], Youn Na[2], Jonathan A Cooper[2], Yves Jossin[1]*

[1]Laboratory of Mammalian Development & Cell Biology, Institute of Neuroscience, Université Catholique de Louvain, Brussels, Belgium; [2]Division of Basic Sciences, Fred Hutchinson Cancer Research Center, Seattle, United States

**Abstract** The functions of FGF receptors (FGFRs) in early development of the cerebral cortex are well established. Their functions in the migration of neocortical projection neurons, however, are unclear. We have found that FGFRs regulate multipolar neuron orientation and the morphological change into bipolar cells necessary to enter the cortical plate. Mechanistically, our results suggest that FGFRs are activated by N-Cadherin. N-Cadherin cell-autonomously binds FGFRs and inhibits FGFR K27- and K29-linked polyubiquitination and lysosomal degradation. Accordingly, FGFRs accumulate and stimulate prolonged Erk1/2 phosphorylation. Neurons inhibited for Erk1/2 are stalled in the multipolar zone. Moreover, Reelin, a secreted protein regulating neuronal positioning, prevents FGFR degradation through N-Cadherin, causing Erk1/2 phosphorylation. These findings reveal novel functions for FGFRs in cortical projection neuron migration, suggest a physiological role for FGFR and N-Cadherin interaction in vivo and identify Reelin as an extracellular upstream regulator and Erk1/2 as downstream effectors of FGFRs during neuron migration.
DOI: https://doi.org/10.7554/eLife.47673.001

*For correspondence:
yves.jossin@uclouvain.be

## Introduction

The mammalian neocortex has a complicated structure, with multiple layers of different types of neurons linked together in microcircuits. Cortical neurons are generated by progenitor divisions at or near the ventricular zone (VZ), then migrate to defined positions in the developing cortical plate (CP) where they differentiate and connect among themselves and to other brain regions. In general, projection neurons arise from the local VZ, below the CP, while interneurons migrate tangentially from the ganglionic eminences (*Parnavelas, 2000*; *Rash and Grove, 2006*). The projection neurons are layered in birth order, such that neurons born early in development are below those born later (*Takahashi et al., 1999*). Correct neuron layering is important: major defects in projection neuron migration may underlie lissencephalies and heterotopias coupled with mental retardation while milder defects are associated with mental disabilities including dyslexia, schizophrenia, epilepsy, and autism spectrum disorders (*Bielas et al., 2004*; *Francis et al., 2006*; *Romero et al., 2018*). A great deal is known about the distinct neuron migration paths and the signals that regulate them but a significant number of patients with developmental disorders lack a diagnosis. It is thus crucial to identify further causes of disease and understand them mechanistically.

Migrating projection neurons have been visualized in fetal mouse brains in which individual neurons have been labeled in utero (*Tabata and Nakajima, 2001*; *Nadarajah and Parnavelas, 2002*; *Noctor et al., 2004*). Migration occurs in distinct phases. First, neurons move radially from the VZ into the lower intermediate zone (IZ). They then become multipolar and despite an apparent

irregular movement, move towards the CP (*Nadarajah et al., 2003*; *Tabata and Nakajima, 2003*; *Noctor et al., 2004*; *Jossin and Cooper, 2011*). When multipolar neurons reach the upper IZ, they become bipolar and traverse the CP by glial-guided locomotion. The lower IZ therefore constitutes a multipolar migration zone (MMZ) and the upper IZ and CP a radial migration zone (RMZ). Movement out of the MMZ is regulated by many signals, which, if disrupted, can lead to altered layering (*Cooper, 2014*; *Kon et al., 2017*). One signal that regulates exit from the MMZ is Reelin, an extracellular matrix protein that activates the small GTPases Rap1A and RalA to upregulate surface levels of neural cadherin (NCad, CDH2) (*Jossin, 2011*; *Jossin and Cooper, 2011*). Inhibiting Reelin, Rap1 or NCad interferes with the orientation of multipolar neurons, increases sideways and downwards movements, and increases time spent in the MMZ. Defects in multipolar migration may contribute to the altered layering in Reelin mutant mice and humans (*Lambert de Rouvroit and Goffinet, 1998*; *Hong et al., 2000*).

Fibroblast growth factors (FGFs) and their receptors (FGFRs) are important during the development of many tissues and for wound healing, tissue repair and metabolism after birth (*Beenken and Mohammadi, 2009*; *Ornitz and Itoh, 2015*). There is wide functional redundancy between family members, with specificity conferred by cell-type-specific expression of FGFs and FGFRs and alternative splicing of FGFRs. At the cellular level, FGFRs regulate cell proliferation, migration, differentiation, survival and cell shape. At the molecular level, FGF and heparan sulfate proteoglycan bind to FGFRs and induce FGFR dimerization and activation. Activated FGFRs autophosphorylate on multiple cytoplasmic tyrosine residues, followed by recruitment and phosphorylation of a variety of downstream signaling proteins (*Ornitz and Itoh, 2015*). Following activation, FGFRs are down-regulated by ubiquitination, endocytosis and lysosomal degradation (*Katzmann et al., 2002*; *Haugsten et al., 2005*). Defects in FGFR activation or down-regulation can lead to anomalous signaling and are associated with developmental defects, metabolic disorders and cancer (*Wesche et al., 2011*; *Ornitz and Itoh, 2015*).

In addition to FGFs and heparan sulfate, cell surface proteins including neural cadherin (NCad or CDH2), epidermal cadherin (ECad or CDH1), L1 cell adhesion molecule (L1CAM) and neural CAM (NCAM) can also bind to and activate FGFRs (*Williams et al., 1994*; *Williams et al., 2001*; *Suyama et al., 2002*; *Brown et al., 2016*). Both NCad and FGFRs are highly expressed during the epithelial-mesenchymal transition of cancer cells and their interaction may be important for metastasis. Indeed, tumor cells artificially over-expressing NCad require FGFR activity for metastasis. Nevertheless, the role of NCad-dependent FGFR activation in metastasis pathology remains unclear (*Hulit et al., 2007*; *Qian et al., 2014*).

In the developing cerebral cortex FGFR1-3 are expressed in the VZ and MMZ (*Iwata and Hevner, 2009*; *Hébert, 2011*) (Allen Institute for Brain Science. Allen Developing Mouse Brain Atlas. Available from: http://developingmouse.brain-map.org/) and FGFRs have been associated with neurodevelopmental diseases including schizophrenia (*O'Donovan et al., 2009*; *Terwisscha van Scheltinga et al., 2013*), epilepsy (*Coci et al., 2017*; *Okazaki et al., 2017*), autism spectrum disorders (*Wentz et al., 2014*; *Coci et al., 2017*) and lissencephaly (*Tan and Mankad, 2018*), suggesting possible roles in neuron migration. However, analysis of cortical neuron migration in FGFR mutant mice has been inconclusive for two reasons. First, functional redundancy may suppress the phenotypes of loss of function mutants (*Beenken and Mohammadi, 2009*; *Hébert, 2011*; *Ornitz and Itoh, 2015*). Second, FGFRs are needed for developmental steps that occur before migration, such as regional patterning of the cortex, neurogenesis and radial glia differentiation (*Shin et al., 2004*; *Rash and Grove, 2006*; *Mason, 2007*; *Kang et al., 2009*; *Paek et al., 2009*). Therefore, the roles of FGFRs in neocortical neuron migration are unclear.

In this paper we report that FGFR1-3 have overlapping functions during the multipolar migration in vivo. FGFRs are required downstream of Rap1 for multipolar cells to orient towards the CP, adopt bipolar morphology, and migrate out of the MMZ. We found that Rap1-dependent NCad upregulation stabilizes FGFRs by inhibiting K27- and K29-linked polyubiquitination and lysosomal degradation and that NCad-FGFR *cis* interaction (on the same cell) is involved. Consequently, FGFRs accumulate and are activated, resulting in prolonged activation of Erk1/2 when neurons are stimulated in vitro with Reelin. In vivo inhibition of K27-linked polyubiquitination or overexpression of FGFRs rescues the migration of neurons with inhibited Rap1. Inhibition of Erk1/2 activity in the developing cerebral cortex induces a similar phenotype as FGFR or Rap1 inhibition. These data reveal a novel function of FGFRs in cortical projection neuron migration and the control of its activity

by ubiquitination and NCad interaction in vivo. To our knowledge, this is the first physiological role for FGFR-NCad interaction during tissue development. Furthermore, we identified FGFRs as mediating Reelin activation of Erk1/2 to control migration during the multipolar phase. These findings provide insights into FGFR mutation-related inherited brain diseases.

## Results

### FGFRs are required for multipolar neurons to orient correctly and become bipolar in vivo

To avoid potential functional redundancy, we tested the importance of FGFRs in neuron migration by inhibiting all family members. Cytoplasmic domain deletion mutants of FGFR1-3 are dominant negative (DN) because they form non-functional heterodimers with all FGFR family members (*Ueno et al., 1992*). To avoid effects on neurogenesis, DN mutants were expressed from the NeuroD promoter, which is activated after cells leave the VZ (*Jossin and Cooper, 2011*). Apical neural stem cells located at the VZ were electroporated in utero (*Tabata and Nakajima, 2001*) at embryonic day E14.5 with DN FGFR1-3 along with GFP and the positions of daughter cells were monitored 3 days later at E17.5. While most control neurons expressing GFP alone had entered the RMZ, neurons over-expressing DN mutant but not full-length FGFR1-3 were arrested in the MMZ (*Figure 1a*). These results suggest that the FGFR1-3 cytoplasmic domains are important for multipolar migration. The knock-down of FGFR1 or FGFR2 using specific shRNAs also induced an arrest of cells at the MMZ, with a more pronounced phenotype when the two receptors are downregulated together (*Figure 1b*, *Figure 1—figure supplement 1*). The knock-down of FGFR3 resulted in a small, statistically non-significant effect on cell positioning (*Figure 1b*, *Figure 1—figure supplement 1*). These results suggest that FGFRs work redundantly with a prominent role for FGFR1 and FGFR2.

To test whether inhibition of FGFRs alters cell proliferation, fate or apoptosis, we examined marker expression 2 days after electroporation. At this stage there is no significant difference between FGFR-inhibited and control cells in their position in the cortex, with most GFP+ cells located within the IZ. FGFR1(DN) had no effect on the proportion of GFP+Ki67+ proliferative cells, GFP+Sox2+ apical neural stem cells or GFP+Tbr2+ basal progenitors (*Figure 2a*). FGFR-inhibited neurons were correctly specified, as shown by the normal expression of Satb2, a marker for upper layer neurons born at the time of the electroporation. Immunostaining for activated caspase-3 showed no increase in cell death (*Figure 2a*).

To gain insight into the mechanism underlying the migration defect, we analyzed the morphology of migrating neurons. Analysis of the morphology revealed no difference in the number of neurites or in the cell body length-to-width ratio of FGFR-inhibited multipolar neurons compared to control multipolar neurons (*Figure 2b–d*). However, while most control multipolar cells had their Golgi apparatus oriented towards the CP, fewer FGFR-inhibited neurons had their Golgi facing the CP, suggesting a failure to orient correctly (*Figure 2e*). In addition, while most control electroporated neurons at the multipolar to radial transition zone had transformed into bipolar cells, FGFR-inhibited neurons were still mostly multipolar (*Figure 2f*). Yet, the few FGFR-inhibited bipolar neurons migrating in the RMZ exhibited no difference in the length of the leading process and the cell body length-to-width ratio compared to control cells and possess an axon at the rear (*Figure 2g–i*). These results suggest that FGFRs are required for multipolar neurons to orient correctly, become bipolar, exit the MMZ, and enter the RMZ. For simplicity we will call this phenotype a defect in multipolar migration.

### Rap1 and NCad regulate FGFRs protein levels to control multipolar migrating neurons in vivo

Since the phenotype induced by dominant-negative FGFRs resembles that induced by inhibiting Reelin receptors, NCad or Rap1 (*Jossin and Cooper, 2011*), there may be a common mechanism. Therefore, we tested for epistasis by over-expressing FGFRs when Rap1 is inhibited by the Rap1 GTPase-activating protein (Rap1GAP). The migration defect induced by Rap1GAP was partly suppressed by over-expression of wild-type FGFR1, 2 or 3 (*Figure 3a*). This suggests that signals from the Reelin-Rap1-NCad pathway may require FGFRs to stimulate multipolar migration, perhaps in parallel with or downstream of NCad.

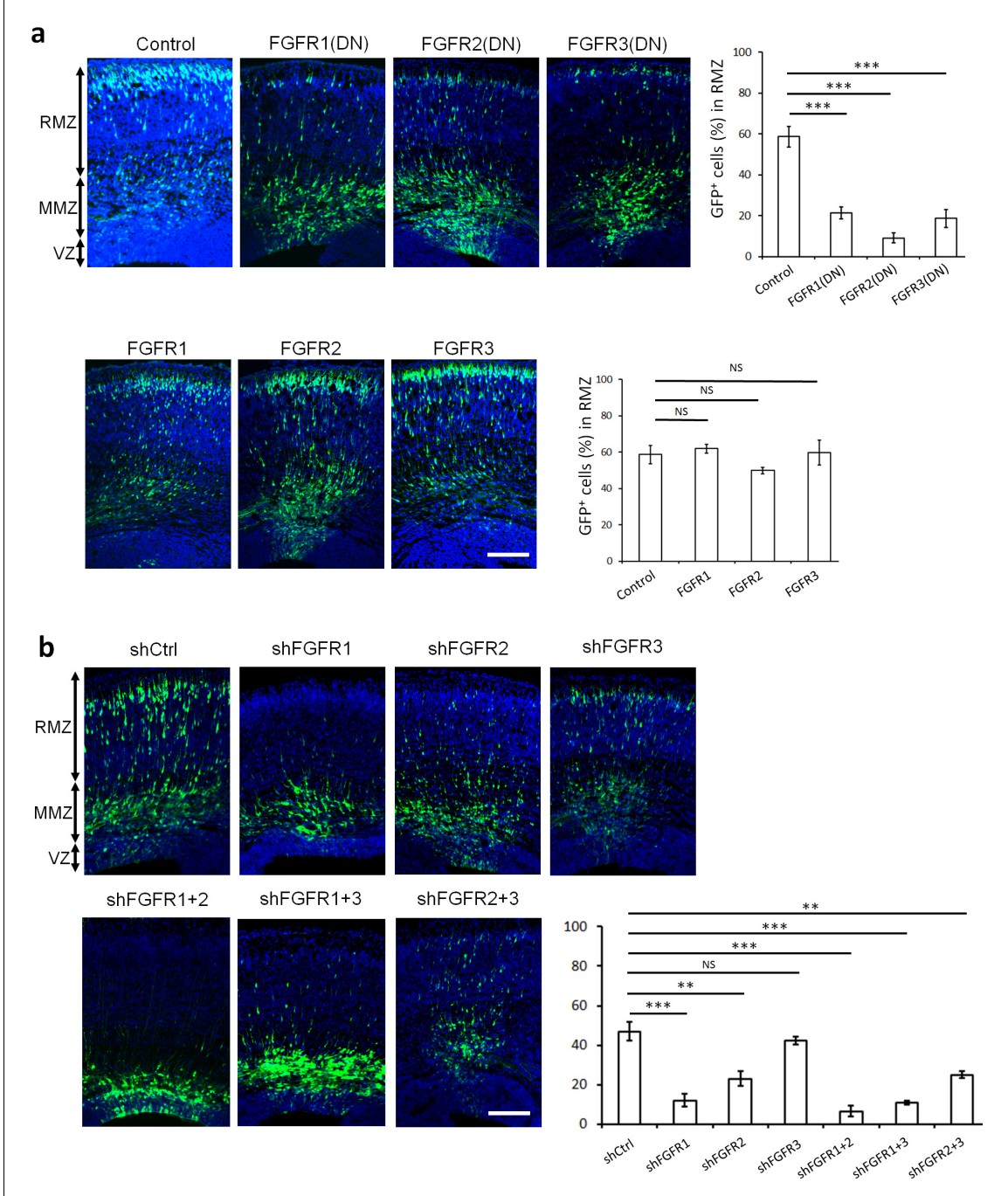

**Figure 1.** FGFRs regulate projection neuron migration in vivo. (a,b) shRNA against FGFR1, 2 or 3 and dominant-negative (DN) but not wildtype FGFR1, 2 or 3 induce an accumulation of neurons at the MMZ. (a) pNeuroD-FGFR(DN) and pNeuroD-FGFR plasmids, expressed in neurons, and (b) shRNA against FGFR1, 2 or 3 or negative control shRNA (shCtrl) were co-electroporated in utero with pCAG-GFP, expressed in progenitors and neurons, at embryonic day (E) 14.5. (a,b) Three days later, cryosections were prepared and labeled for DAPI (blue) and GFP (green). The cerebral wall was subdivided into radial morphology zone (RMZ), multipolar morphology zone (MMZ) and VZ. Graphs show the percentage of cells in the RMZ. *n* = 5 Control, 7 FGFR1(DN), 4 FGFR2(DN), 4 FGFR3(DN), 3 FGFR1, 3 FGFR2, 4 FGFR3, 6 shCtrl, 4 shFGFR1, 4 shFGFR2, 4 shFGFR3, 5 shFGFR1+2, 3 shFGFR1+3, 3 shFGFR2+3. *P* values: FGFR1(DN): 9.6E-6, FGFR2(DN): 4.2E-6, FGFR3(DN): 4.0E-5, FGFR1: 0.245, FGFR2: 0.170, FGFR3: 0.353, shFGFR1: 3.0E-4, shFGFR2: 2.9E-3, shFGFR3: 0.169, shFGFR1+2: 6.4E-5, shFGFR1+3: 1.0E-4, shFGFR2+3: 3.2E-3. Error bars, s.e.m. ***p<0.001, **p<0.01, *p<0.05, NS, not significant Scale bar 100 μm.

DOI: https://doi.org/10.7554/eLife.47673.002

The following source data and figure supplement are available for figure 1:

*Figure 1 continued on next page*

*Figure 1 continued*

**Source data 1.** FGFRs regulate projection neuron migration in vivo.
DOI: https://doi.org/10.7554/eLife.47673.004
**Figure supplement 1.** Efficiency of shRNAs against FGFR1, 2, 3.
DOI: https://doi.org/10.7554/eLife.47673.003

NCad can bind to, stabilize, and activate FGFR1 in cell culture (*Suyama et al., 2002*; *Sanchez-Heras et al., 2006*), providing a potential mechanism for FGFR activation in multipolar neurons. Therefore, we tested whether Rap1 regulates FGFR protein abundance in vivo. Since we could not reliably detect endogenous FGFR by immunofluorescence, we co-electroporated FGFR1-GFP and CherryFP with Rap1GAP or control plasmid in utero. The level of FGFR1-GFP in CherryFP+ neurons was reduced when Rap1 was inhibited (*Figure 3b*). In addition, over-expressing NCad-HA in cultured primary neurons increased the level of endogenous FGFR1 (*Figure 3c*), consistent with NCad mediating Rap1-dependent FGFR1 stabilization in vivo. As expected, expressing NCad but not ECad increased protein levels of all three FGFRs in cultured cells (*Figure 3—figure supplement 1*). Control experiments showed that FGFR inhibition did not change the protein abundance of NCad and did not perturb NCad homophilic interaction properties (*Figure 3—figure supplement 1b*). NCad was still able to accumulate at cell-cell junctions in the presence of FGFR1(DN) (*Figure 3—figure supplement 1c*).

These results extend previous reports that NCad can increase FGFRs protein levels in cell culture to an in vivo developmental system.

## NCad homophilic adhesion is dispensable for the multipolar migration and for increasing FGFR protein levels

If NCad regulates FGFRs during multipolar migration, NCad-mediated cell-cell adhesion may be dispensable. To test this possibility, we generated a mutant NCad that is incapable of forming homophilic cell-cell adhesion. W161 of NCad (numbered from the initiator methionine, corresponding to W2 in the mature protein) is required for NCad-NCad binding between cells (*Tamura et al., 1998*; *Pertz et al., 1999*). As expected, NCad[W161A] did not bind NCad expressed on different cells (*Figure 4—figure supplement 1a* lane 2) but still bound NCad expressed on the same cell (*Figure 4—figure supplement 1b* lane 2). Remarkably, NCad[W161A] rescued the movement of Rap1-inhibited neurons in vivo (*Figure 4*), and increased FGFR1 protein level in vitro (*Figure 4—figure supplement 1c*). NCad[W161A] was expressed at the same level as NCad (*Figure 4—figure supplement 2*). These results suggest that NCad function in multipolar migration is independent of NCad-NCad *trans* interactions but may require NCad binding to FGFR.

## NCad EC4 is required for NCad-FGFR Cis interaction and multipolar migration in vivo

To test whether NCad-FGFR binding is necessary to increase FGFR protein levels and rescue migration, we generated an NCad mutant that does not bind FGFRs. To do this, we deleted NCad extracellular domain 4 (EC4), previously reported to mediate NCad-FGFR binding (*Williams et al., 2001*). NCad[ΔEC4] no longer bound to FGFR1 in transfected cells (*Figure 5a*), although it retained homophilic binding to co-transfected NCad (*Figure 5b*). Also, NCad[ΔEC4] failed to increase the protein abundance of co-transfected FGFR1 or to activate FGFRs, as observed by an increase in FGFR auto-phosphorylation on tyrosines 653/654 and phosphorylation of the well-known downstream signaling kinases Erk1/2 (*Figure 5c*). Finally, an FGFR inhibitor (*Nakanishi et al., 2014*) prevented NCad-induced FGFR auto-phosphorylation and phosphorylation of Erk1/2 (*Figure 5c*). We also found that NCad binds FGFRs in *cis*, on the same cell, but not *trans*, between cells (*Figure 5—figure supplement 1*). Together, these results show that NCad *cis* interaction with FGFRs induces FGFR accumulation and FGFR-dependent Erk1/2 phosphorylation in cell culture. Importantly, NCad[ΔEC4] did not rescue the migration of Rap1-inhibited neurons (*Figure 5d*). The requirement for EC4 to bind and activate FGFRs and to rescue migration supports the idea that cell autonomous NCad-FGFR binding and activation are required to stimulate multipolar migration in vivo.

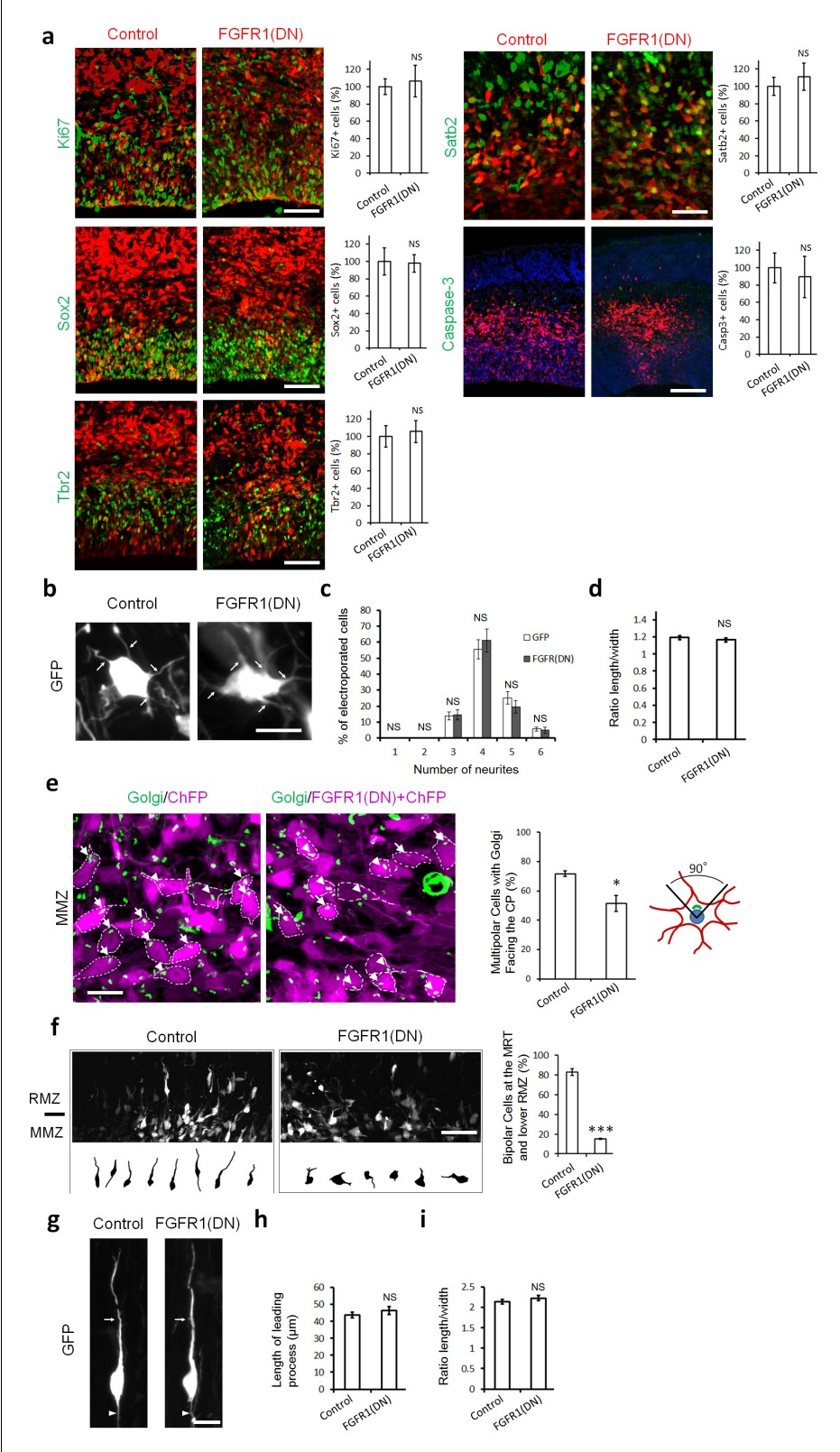

**Figure 2.** Inhibiting FGFRs in post-mitotic neurons has no effect on proliferation and differentiation but regulates multipolar neuron orientation and morphology. In utero electroporation was performed at embryonic day E14.5 and analyzed 2 days later (**a–f**) or 3 days later (**g–i**). (**a**) Inhibition of FGFRs did not affect cell division (Ki67), apical (Sox2) or basal (Tbr2) progenitor cells, neuronal commitment (Satb2), or survival (cleaved Caspase-3). Expression of CherryFP (red) alone (control) or with FGFR1(DN) as indicated. After immunostaining for the indicated markers (green), the results were quantified by

*Figure 2 continued on next page*

Figure 2 continued

counting the number of labeled electroporated cells in a constant area of each section and averaged across sections from at least three different embryos for each antibody. Values are normalized to control (100%). (mean ± s.e.m.). NS, not significant. Scale bars, 50 µm for Ki67, Sox2 and Tbr2, 25 µm for Satb2, 100 µm for cleaved Caspase 3. (b, c, d) Inhibition of FGFR did not affect the number of neurites or the length to width morphology of multipolar cells. (b) High magnification of GFP+ multipolar neurons within the MMZ following overexpression of GFP or FGFR(DN). (c) Proportion of GFP+ cells with the indicated number of neurites within the MMZ. (d) Ratio of length/width of the GFP+ cells within the MMZ as an indicator of cell shape. *P* value: 0.196. (mean ± s.e.m.). NS, not significant. Arrows indicate the neurites, arrowheads indicate the axons. Scale bar 10 µm e) FGFR-inhibited neurons are disoriented. Golgi staining (green) of MMZ neurons (purple). The figure shows examples of multipolar neurons with their Golgi facing the CP (white arrows) or facing other directions (white arrowheads). The percentage of cells with Golgi facing the cortical plate was calculated (mean ± s.e.m.). *p<0.05, *P* value: 0.013. Scale bar 10 µm. (f) FGFR inhibition affects the multipolar to radial transition. Computer-based reconstruction of GFP+ neurons morphologies at the multipolar to radial transition zone (MRT) and the lower RMZ. The graph shows the percentage of bipolar radially oriented neurons. Scale bar 30 µm. Error bars, s.e.m. ***p<0.001, *P* value: 6.5E-6. (g, h, i) Inhibition of FGFR did not affect the length of the leading process and the length-to-width morphology of radially migrating cells. (g) High magnification of GFP+ bipolar neurons within the RMZ following overexpression of GFP or FGFR(DN). (h) Length of the leading process of GFP+ bipolar cells within the RMZ. *P* value: 0.180. (i) Ratio of length/width of the GFP+ cells within the RMZ as an indicator of cell shape. *P* value: 0.155 Arrows indicate the leading process, Scale bar 10 µm. (mean ± s.e.m.). NS, not significant.

DOI: https://doi.org/10.7554/eLife.47673.005

The following source data is available for figure 2:

**Source data 1.** Inhibiting FGFRs in post-mitotic neurons has no effect on proliferation and differentiation but regulates multipolar neuron orientation and morphology.
DOI: https://doi.org/10.7554/eLife.47673.006

## NCad but not ECad domains 1–2 increase FGFR protein levels and promote multipolar migration in vivo

Cell culture studies differ as to whether FGFRs bind to both NCad and ECad or only NCad (*Williams et al., 2001*; *Brown et al., 2016*). Since we found that NCad but not ECad rescues multipolar migration in vivo (*Figure 4a*), despite being expressed at similar levels (*Figure 4—figure supplement 2*), we re-examined interactions between FGFRs, NCad and ECad in cell culture. In our hands, NCad, NCad^W161A and ECad all bound FGFRs in transfected cells (*Figure 6—figure supplement 1*) but only NCad and NCad^W161A increased the protein levels of co-transfected FGFRs (*Figure 3—figure supplement 1a* and *Figure 4—figure supplement 1c*). Thus, NCad but not ECad rescue of multipolar migration correlates with the ability to increase FGFR protein abundance while receptors interaction is necessary but not sufficient.

Our results suggest that a unique feature of NCad, not shared with ECad, is required to increase FGFR protein levels and stimulate migration. We identified the critical region of NCad by the use of NCad/ECad chimeras. Classic cadherins are composed of an extracellular domain (ECD) with five extracellular cadherin (EC) repeats and a highly conserved intracellular domain (ICD) that interacts with signaling proteins. We switched the entire ECD and ICD of NCad and ECad, creating ENCad and NECad (*Figure 6a*). NECad but not ENCad was able to increase FGFR protein abundance (*Figure 6a*). Moreover, NECad but not ENCad rescued the positional defect observed when Rap1 is inhibited in vivo (*Figure 6b*). This suggests that the specificity of NCad to protect FGFR from degradation and to function in multipolar migration in vivo lies in the ECD. To map the NCad-specific function more closely, we prepared two other chimeric proteins: ENNCad has the EC1, EC2 and the first half of EC3 domains of ECad and the remainder of NCad, while NENCad has the second half of EC3, EC4 and EC5 domains of ECad and the remainder of NCad. NENCad but not ENNCad was able to increase FGFR protein level and rescue the positional defect observed when Rap1 is inhibited in vivo (*Figure 6*). These results show that EC4-5 of either NCad or ECad can protect FGFRs from degradation and stimulate migration provided that EC1-2 and part of EC3 are derived from NCad. Overall, these data demonstrate that cadherin EC4 interaction with FGFR is necessary but not sufficient to stabilize and activate FGFRs and regulate multipolar migration in vivo. Additional unique features carried by NCad EC1-2 are also needed.

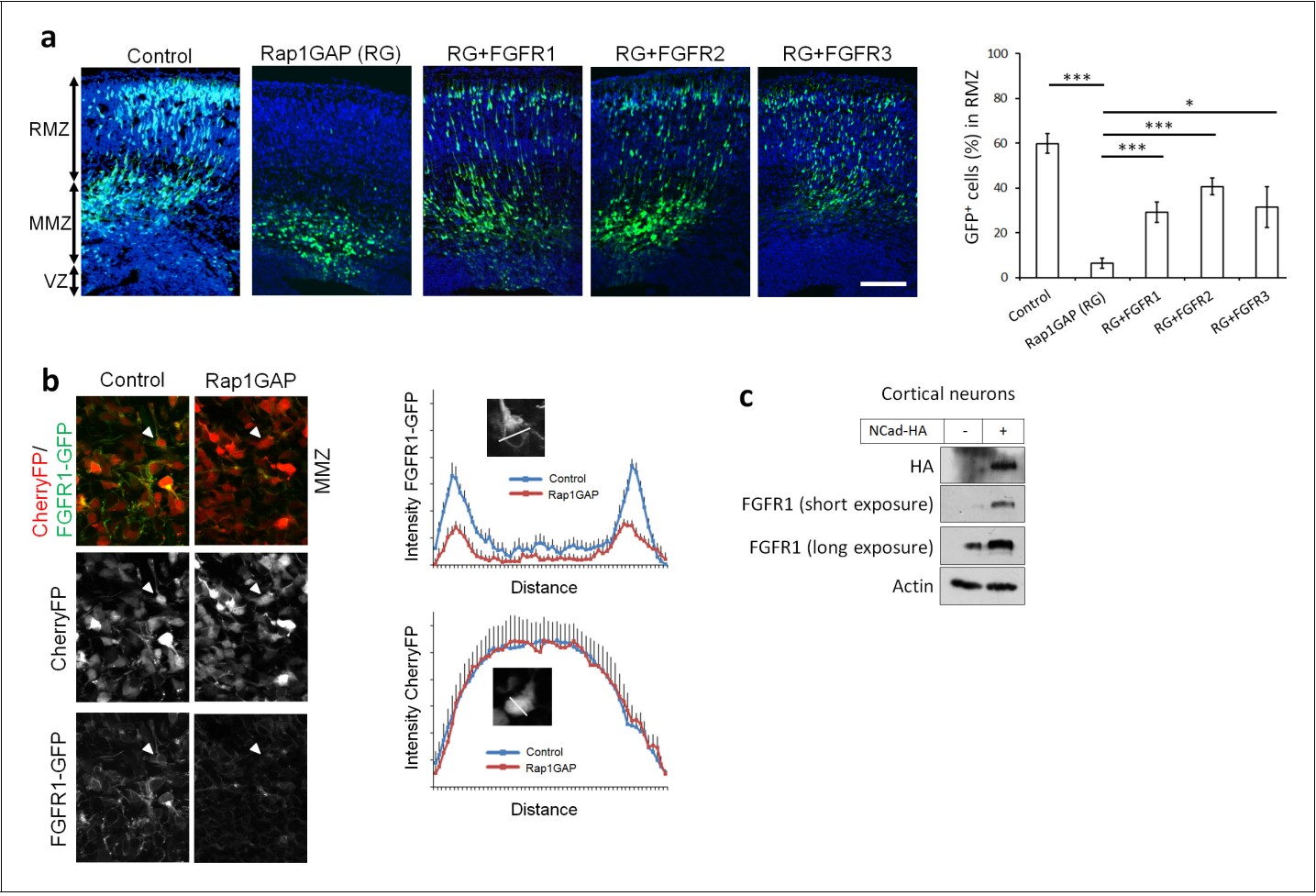

**Figure 3.** Rap1 and NCad regulate FGFR levels and function in multipolar migrating neurons. (a) FGFR1, 2 and 3 partially rescue the neuronal migration phenotype induced by Rap1 inhibition. E14.5 embryos were electroporated in utero with pCAG-GFP, pNeuroD vector or pNeuroD-Rap1GAP (RG), and pNeuroD-FGFR1, 2 or 3 as shown. Cryosections were prepared 3 days later and labeled for DAPI (blue) and GFP (green). The cerebral wall was subdivided into radial morphology zone (RMZ), multipolar morphology zone (MMZ) and VZ. Graphs show the percentage of cells in the RMZ (mean ± s. e.m.). ***p<0.001; *p<0.05, P values: Rap1GAP (RG): 9.8E-8, RG+FGFR1: 7.0E-4, RG+FGFR2: 3.0E-4, RG+FGFR3: 0.020 (n = 4 Control, 4 Rap1GAP (RG), 7 RG+FGFR1, 7 RG+FGFR2, 4 RG+FGFR3). (b) Protein abundance of FGFR1-GFP is regulated by Rap1 in vivo. E14.5 embryos were electroporated in utero with a mixture of pCAG-CherryFP, pNeuroD-FGFR1-GFP and either vector or pNeuroD-Rap1GAP. Two days later, mCherry and FGFR1-GFP were detected by epifluorescence. The graphs show mean and standard deviation of image intensity measured across lines drawn through the center of the cell body for eight neurons in each case. (c) Embryonic cortical neurons were electroporated to overexpress pCAG-NCad-HA or with a control plasmid, cultured for 2 days then analyzed for the protein level of NCad-HA and endogenous FGFR1 by Western blot.
DOI: https://doi.org/10.7554/eLife.47673.007

The following source data and figure supplement are available for figure 3:

**Source data 1.** FGFR1, 2 and 3 partially rescue the neuronal migration phenotype induced by Rap1 inhibition.
DOI: https://doi.org/10.7554/eLife.47673.009
**Figure supplement 1.** FGFR protein levels are increased by NCad but not by ECad.
DOI: https://doi.org/10.7554/eLife.47673.008

## FGFR K27/K29-linked polyubiquitination and lysosomal degradation control multipolar migration in vivo

The Rap1/NCad-dependent increase in FGFR protein in vivo (*Figure 3b,c*) and in vitro (*Figure 3— figure supplement 1a*) suggests that NCad may inhibit FGFR degradation, as observed in some cancer cells (*Suyama et al., 2002*). Degradation of many cell surface receptors involves ubiquitination of their cytoplasmic domains and targeting to the lysosome (*Piper et al., 2014*). When FGFR1-Myc was co-expressed with HA-ubiquitin, a ladder of FGFR1-ubiquitin conjugates could be

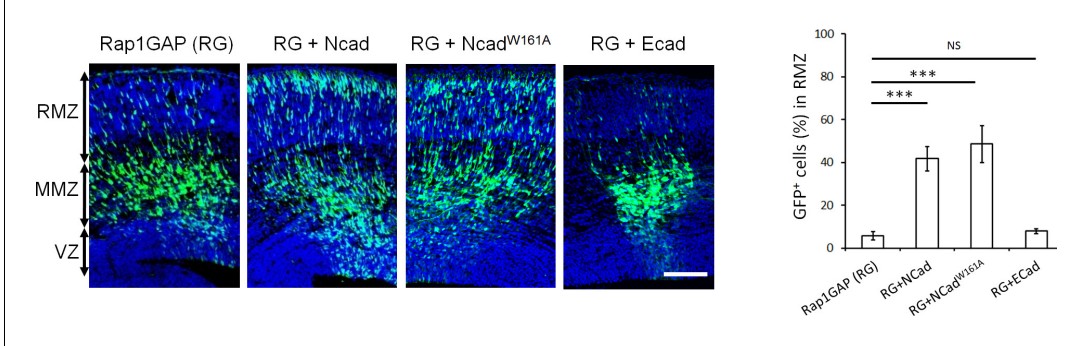

**Figure 4.** NCad homophilic binding mutant NCad[W161A] but not ECad rescues multipolar migration of Rap1-inhibited neurons. E14.5 embryos were electroporated in utero with pCAG-GFP, pNeuroD-Rap1GAP (RG), and pNeuroD vector, NCad, NCad[W161A] or ECad. Cryosections were prepared 3 days later and labeled for DAPI (blue) or GFP (green). The graph shows the percentage of cells in the RMZ (mean ± s.e.m.). *P* values: RG+NCad: 9.8E-6, RG+ NCad[W161A]: 9.3E-6, RG+ECad: 0.213 (*n* = 5 Rap1GAP (RG), 5 RG+NCad, 6 RG+NCad[W161A], 5 RG+ECad). Error bars, s.e.m. ***p<0.001, NS, not significant. Scale bar 100 μm.

DOI: https://doi.org/10.7554/eLife.47673.010

The following source data and figure supplements are available for figure 4:

**Source data 1.** NCad homophilic binding mutant NCad[W161A] but not ECad rescues multipolar migration of Rap1-inhibited neurons.
DOI: https://doi.org/10.7554/eLife.47673.013
**Figure supplement 1.** NCad W161A mutation inhibits trans but not cis homophilic binding.
DOI: https://doi.org/10.7554/eLife.47673.011
**Figure supplement 2.** Ncad-HA, NcadW161A-HA and ECad-HA are expressed at equal levels by in utero electroporation.
DOI: https://doi.org/10.7554/eLife.47673.012

immunoprecipitated with antibodies against Myc (*Figure 7a* lane 3) or HA (data not shown). FGFR ubiquitination was inhibited by co-expressed NCad but not NCad[ΔEC4] (*Figure 7a*) or ECad (data not shown), consistent with NCad binding inhibiting FGFR ubiquitination. Ubiquitin ladders can result from the addition of single ubiquitin moieties at many sites (multi-monoubiquitination) or addition of ubiquitin chains (polyubiquitination). Seven lysine (K) residues on the ubiquitin molecule may be used for polyubiquitination, resulting in diverse outcomes for the target protein (*Ikeda and Kerppola, 2008*; *Fushman and Wilkinson, 2011*; *Sadowski et al., 2012*). To test whether over-expressed FGFR1-Myc is multi-monoubiquitinated or polyubiquitinated, we inhibited polyubiquitination by co-over-expressing a ubiquitin mutant (Ubi[K0]), in which all 7 K residues substituted to arginine (R) (*Lim et al., 2005*). When co-expressed with FGFR1, HA-Ubi[K0] but not HA-Ubi[WT] increased FGFR1 protein level and decreased FGFR1 ubiquitination (*Figure 7b*) suggesting polyubiquitination. To identify the specific polyubiquitin linkage, we used ubiquitin mutants that contain single K to R substitutions. We found that over-expressing Ubi[K27R] or Ubi[K29R] but not other mutants increased FGFR protein level to the same level as that induced by the presence of NCad-HA (*Figure 7c*). Co-expression of Ubi[K27R] or Ubi[K29R] with NCad did not have any cumulative effect, suggesting that FGFR is degraded following attachment of K27- and K29-linked polyubiquitin and that NCad inhibits this process. To test whether K27 or K29 linkages are sufficient or whether both are needed, we used ubiquitin mutants where all lysines except 27 or 29 are mutated to arginine (Ubi[K27] and Ubi[K29]). Co-expressing together Ubi[K27] and Ubi[K29], which allows only K27 and K29 linkages, did not inhibit FGFR degradation (compare to Ubi[WT], *Figure 7d*). However, expressing either Ubi[K27] or Ubi[K29] inhibited FGFR degradation, suggesting that both K27 and K29 ubiquitin linkages are required for FGFR degradation. Importantly, preventing FGFR degradation in vivo by the overexpressing Ubi[K27R] in utero partially rescued the multipolar migration of Rap1-inhibited neurons (*Figure 7e*).

The requirement for both K27 and K29 polyubiquitin linkages suggested that FGFR may be targeted for lysosomal degradation (*Chastagner et al., 2006*; *Ikeda and Kerppola, 2008*; *Zotti et al., 2011*). Indeed, adding lysosome inhibitor leupeptin but not proteasome inhibitor epoxomycin to primary cortical embryonic neurons increased the protein abundance of endogenous FGFR1 (*Figure 7f*). Leupeptin but not epoxomycin also increased levels of transfected FGFR1 to the same extent as over-expressed NCad, suggesting that NCad protects FGFRs from lysosomal degradation

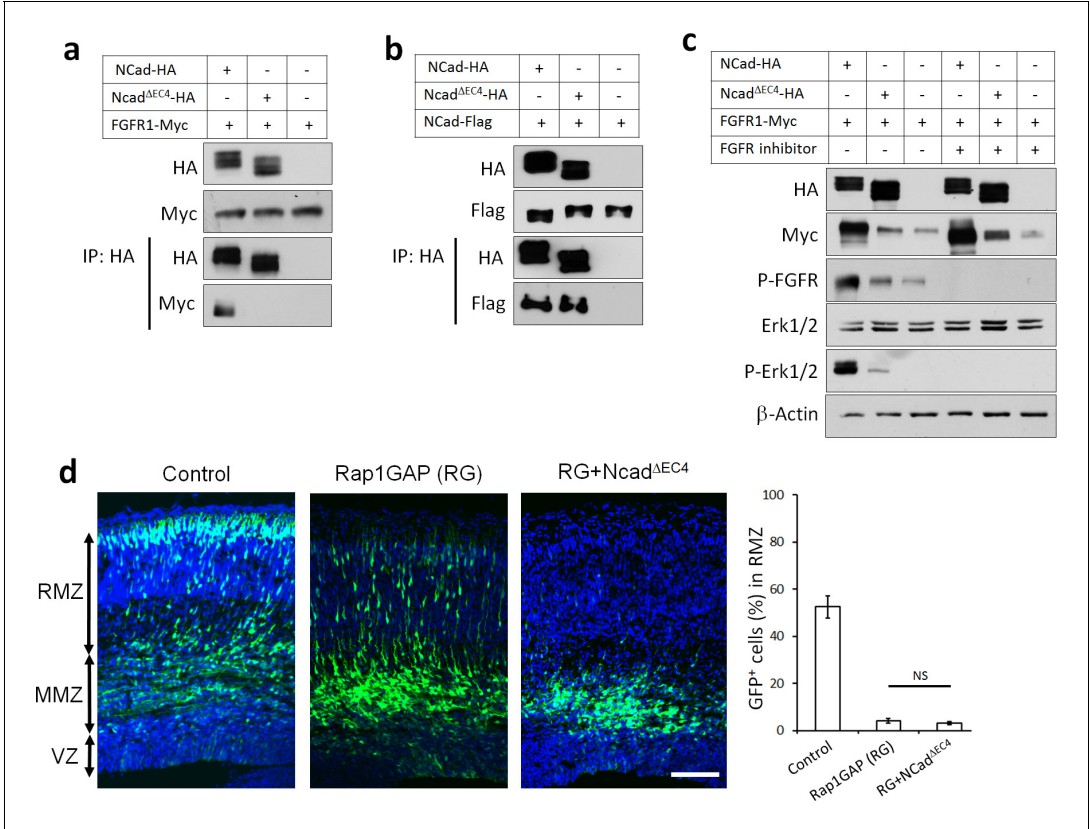

**Figure 5.** NCad-FGFR *cis* interaction through NCad EC4 is required for multipolar migration in vivo. (**a**) NCad EC4 is required for FGFR1 binding in vitro. Cells were transfected with pCAG-FGFR1-Myc and pCAG-NCad-HA, NCad$^{\Delta EC4}$-HA or vector. To equalize FGFR1-Myc expression, half the amount of FGFR1-Myc was transfected with wildtype NCad. One day later, cells were lysed and immunoprecipitated with anti-HA. Lysates and co-immunoprecipitated proteins were analyzed by Western blot. (**b**) EC4 is dispensable for NCad homophilic binding. Cells were transfected with pCAG-NCad-FLAG and pCAG-NCad-HA, NCad$^{\Delta EC4}$-HA or vector. One day later, cells were lysed and immunoprecipitated with anti-HA. Lysates and co-immunoprecipitated proteins were analyzed by Western blot. (**c**) NCad increases FGFR protein level dependent on EC4, and increases FGFR and Erk1/2 phosphorylation dependent on EC4 and FGFR kinase activity. HEK293T cells were transfected with equal amounts of pCAG-FGFR1-Myc DNA and either pCAG-NCad-HA, pCAG-NCad$^{\Delta EC4}$-HA or vector. 24 hr after transfection, the specific FGFR inhibitor Debio1347 was used at 5 μM for 2 hr. Lysates were analyzed by Western blot using the indicated antibodies. Experiments a–c) were repeated independently three times with similar results. (**d**) NCad EC4 is required for the multipolar migration. E14.5 embryos were electroporated in utero with pCAG-GFP and pNeuroD-Rap1GAP (RG), pNeuroD-NCad$^{\Delta EC4}$-HA or vector. Cryosections were prepared three days later and labeled for DAPI (blue) and GFP (green). The graph shows the percentage of cells in the RMZ. n = 4 control, 4 Rap1GAP (RG), 6 RG+ NCad$^{\Delta EC4}$. *P* value: 0.116. Scale bar 100 μm. Error bars, s.e.m., NS, not significant.

DOI: https://doi.org/10.7554/eLife.47673.014

The following source data and figure supplement are available for figure 5:

**Source data 1.** NCad EC4 is required for the multipolar migration.

DOI: https://doi.org/10.7554/eLife.47673.016

**Figure supplement 1.** FGFR1-NCad binding does not occur in *trans*, between cells but occurs in *cis*, within cells.

DOI: https://doi.org/10.7554/eLife.47673.015

(*Figure 7—figure supplement 1*). Lysosome inhibition did not have a cumulative effect on FGFR1 in cells over-expressing NCad, confirming that NCad protects FGFRs from degradation by the lysosome (*Figure 7—figure supplement 1*). We verified that the proteasome inhibitor was active and induced the accumulation of ubiquitinated proteins (*Figure 7—figure supplement 1*). Overall, these results suggest that NCad regulates multipolar migration in vivo by inhibiting FGFR K27- and K29-linked polyubiquitination and degradation through the lysosome, thereby raising FGFR protein levels. This raises the question of whether FGFR levels are increased when NCad is upregulated by Reelin.

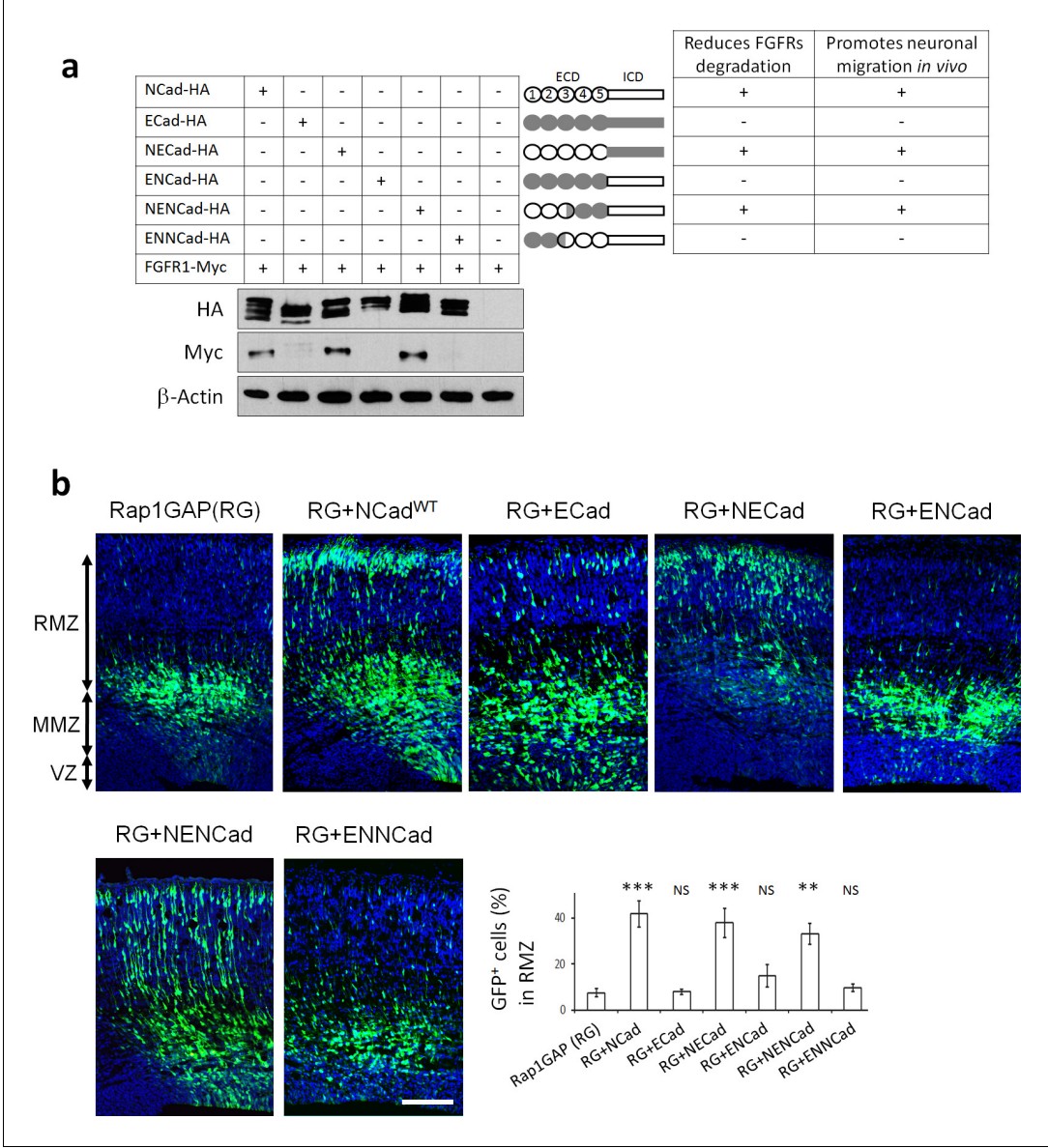

**Figure 6.** NCad EC1-2 are required to increase FGFR protein levels and stimulate multipolar migration in vivo. (a) NCad EC1-2 are necessary to increase FGFR protein abundance. Cells were transfected to express the indicated proteins. 2 days later, protein levels were observed by Western blot. Similar results were obtained from three independent experiments. The figure includes a schematic representing the chimeric proteins used. (b) NCad EC1-2 promote neuronal migration in vivo. In utero electroporation at embryonic day E14.5 and analysis 3 days later. pNeuroD plasmids coding for the indicated proteins and pCAG-GFP were co-electroporated. The graph shows the percentage of cells in the RMZ (mean ± s.e.m.). *P* values: RG+NCad: 9.8E-6, RG+ECad: 0.215, RG +NECad: 5.4E-4, RG+ENCad: 0.080, RG+NENCad: 4.0E-3, RG+ENNCad: 0.206. *n* = 5 Rap1GAP (RG), 5 RG +NCad, 5 RG+ECad, 8 RG+NECad, 5 RG+ENCad, 5 RG+NENCad, 5 RG+ENNCad. Error bars, s.e.m. \*\*\*p<0.001, \*\*p<0.01, NS, not significant.

DOI: https://doi.org/10.7554/eLife.47673.017

The following source data and figure supplement are available for figure 6:

**Source data 1.** NCad EC1-2 promote neuronal migration in vivo.
DOI: https://doi.org/10.7554/eLife.47673.019

**Figure supplement 1.** *FGFR1 binds NCad, NCad^W161A and ECad* HEK293T cells were transfected with FGFR1-Myc and NCad-HA, vector, ECad-HA or NCad^W161A-HA.
DOI: https://doi.org/10.7554/eLife.47673.018

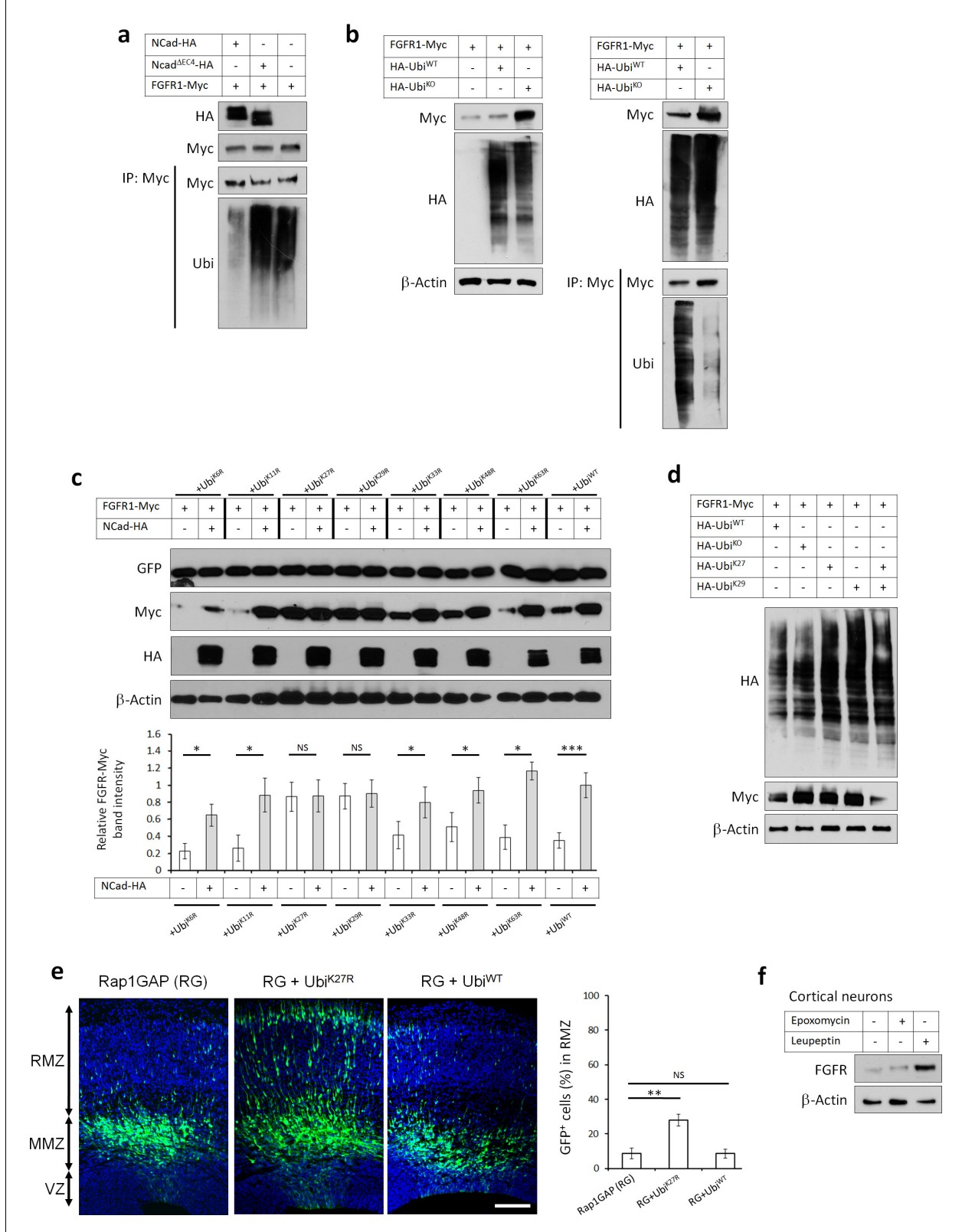

**Figure 7.** FGFRs K27/K29-linked polyubiquitination and lysosomal degradation controls multipolar neuronal migration in vivo. (**a**) NCad but not NCad$^{\Delta EC4}$ inhibits FGFR1 ubiquitination. Cells were transfected with pCAG-FGFR1-Myc and pCAG-NCad, NCad$^{\Delta EC4}$-HA or vector. One day later, cells were lysed and proteins immunoprecipitated with anti-Myc. Lysates and immunoprecipitates were analyzed with Western blotting using antibodies to Myc and ubiquitin. To equalize FGFR1-Myc levels, half the amount of DNA was used for FGFR1-Myc when expressed with NCad-HA. (**b**) HA-Ubi$^{KO}$ but

*Figure 7 continued on next page*

*Figure 7 continued*

not HA-Ubi$^{WT}$ increased FGFR1 protein level and decreased FGFR1 ubiquitination. Cells were transfected with pCAG-FGFR1-Myc and HA-Ubi$^{KO}$ or HA-Ubi$^{WT}$. One day later, cells were lysed and proteins immunoprecipitated with anti-Myc. Lysates and immunoprecipitates were analyzed with Western blotting using antibodies to Myc and HA. (c) Inhibition of K27- and K29-linked polyubiquitination increases FGFR1 protein level. Ubiquitin-GFP mutants in which one lysine is mutated into arginine were used to identify lysine residues required for polyubiquitin chain formation. Co-translational cleavage detaches the GFP and frees the terminal glycine of ubiquitin for subsequent conjugation (*Boname et al., 2010*). The cleaved GFP was used to quantify ubiquitin mutant expression. The graph shows the relative FGFR1-Myc band intensity when expressed in the presence or absence of NCad-HA and in the presence of an ubiquitin mutant as indicated (mean ± s.e.m.). *P* values: Ubi$^{K6R}$: 0.026, Ubi$^{K11R}$: 0.034, Ubi$^{K27R}$: 0.490, Ubi$^{K29R}$: 0.466, Ubi$^{K33R}$: 0.036, Ubi$^{K48R}$: 0.032, Ubi$^{K63R}$: 0.024, Ubi$^{WT}$: 4.8E-4. *n* = 4 Ubi$^{K6R}$, 3 Ubi$^{K11R}$, 3 Ubi$^{K27R}$, 3 Ubi$^{K29R}$, 4 Ubi$^{K33R}$, 4 Ubi$^{K48R}$, 3 Ubi$^{K63R}$, 8 Ubi$^{WT}$. (d) FGFR1 levels remain normal only when both K27- and K29-linked polyubiquitination are permitted. HA-Ubiquitin mutants in which all but one lysine is mutated into arginine were used to allow only one type of polyubiquitin chain formation (Ubi$^{K27}$ and Ubi$^{K29}$). (e) Inhibition of K27-linked polyubiquitin chain formation in vivo rescues the migration defect of Rap1GAP-expressing cells. In utero electroporation at embryonic day E14.5 and analysis 3 days later. Plasmids coding for the indicated proteins and GFP were co-electroporated. The graph shows the percentage of cells in the RMZ (mean ± s.e.m.). *P* values: RG+Ubi$^{K27R}$: 1.7E-3, RG+Ubi$^{WT}$: 0.471. *n* = 4 Rap1GAP (RG), 8 RG+Ubi$^{K27R}$, 6 RG+Ubi$^{WT}$. (f) Endogenous FGFR1 is degraded by the lysosome in vivo. Primary embryonic cortical neurons were cultured in the presence of 250 nM proteasome inhibitor epoxomycin or 300 µM lysosome inhibitor leupeptin for 4 hr and analyzed by Western blot. Similar results were obtained in three independent experiments. Scale bar 100 µm. *p<0.05, **p<0.01 ***p<0.001, NS not significant.

DOI: https://doi.org/10.7554/eLife.47673.020

The following source data and figure supplement are available for figure 7:

**Source data 1.** Inhibition of K27- and K29-linked polyubiquitination increases FGFR1 protein level and rescues the migration defect of Rap1GAP-expressing cells.
DOI: https://doi.org/10.7554/eLife.47673.022

**Figure supplement 1.** NCad over-expression and lysosomal inhibition increase FGFR1 protein level to the same extent.
DOI: https://doi.org/10.7554/eLife.47673.021

## Reelin induces NCad-dependent FGFR and Erk1/2 activation in cortical neurons

To test whether Reelin activation of the Rap1-NCad pathway increases FGFR protein levels and signaling, we stimulated primary cortical embryonic neurons with partly-purified Reelin or Mock conditioned media and assayed FGFR1 levels and Erk1/2 phosphorylation. As a positive control for FGFR1 signaling, neurons were treated for 20 min with FGF2, which activated Erk1/2 phosphorylation (*Figure 8a*). As expected, treating for 20 min with Reelin induced phosphorylation of Dab1, a known rapid effect of Reelin (*Howell et al., 1999*), but did not activate Erk1/2 or increase FGFR1 levels (*Figure 8a*). However, Reelin did increase FGFR1 protein abundance after 15 hr, consistent with inhibition of FGFR degradation and slow accumulation of FGFR1 over time (*Figure 8b* lane 4). Remarkably, 15 hr Reelin treatment also activated Erk1/2, to a similar extent as 20 min exposure to EGF or FGF2 (*Figure 8b* lanes 2 and 3). FGFR1 protein level and Erk1/2 phosphorylation were also increased by 15 hr treatment with R3-6 (*Figure 8c*), a fragment of Reelin that is produced in vivo and is necessary and sufficient for Reelin regulation of neuron migration (*Jossin et al., 2004*; *Jossin et al., 2007*).

Reelin-induced Erk1/2 phosphorylation but not FGFR1 accumulation was completely abrogated by FGFR kinase inhibitor Debio1347. Debio1347 was specific because it inhibited Erk1/2 activation by 20 min FGF2 but did not inhibit Erk1/2 activation by EGF (*Figure 8b*) or Reelin-induced Dab1 phosphorylation (*Figure 8—figure supplement 1*). These results suggest that Reelin-induced Erk1/2 phosphorylation is dependent on FGFR activity and correlates with its effect on FGFR protein levels.

To test whether FGFR1 protein upregulation and Erk1/2 activation require Rap1 or NCad, cortical neurons were transfected with Rap1GAP or NCad$^{DN}$, both of which inhibit NCad accumulation at the plasma membrane and inhibit multipolar migration (*Jossin and Cooper, 2011*). Inhibiting Rap1 or NCad prevented Reelin-induced FGFR1 protein increase and Erk1/2 activation (*Figure 8d*). Taken together, these results show that Reelin or R3-6 can activate FGFR1 signaling and Erk1/2 through the Rap1-NCad pathway.

## Erk1 and Erk2 regulate neuron migration in vivo

Since Reelin, Rap1, NCad and FGFRs are all important to activate Erk1/2 and stimulate multipolar migration, we tested whether Erk1/2 are needed for multipolar migration in vivo. We overexpressed

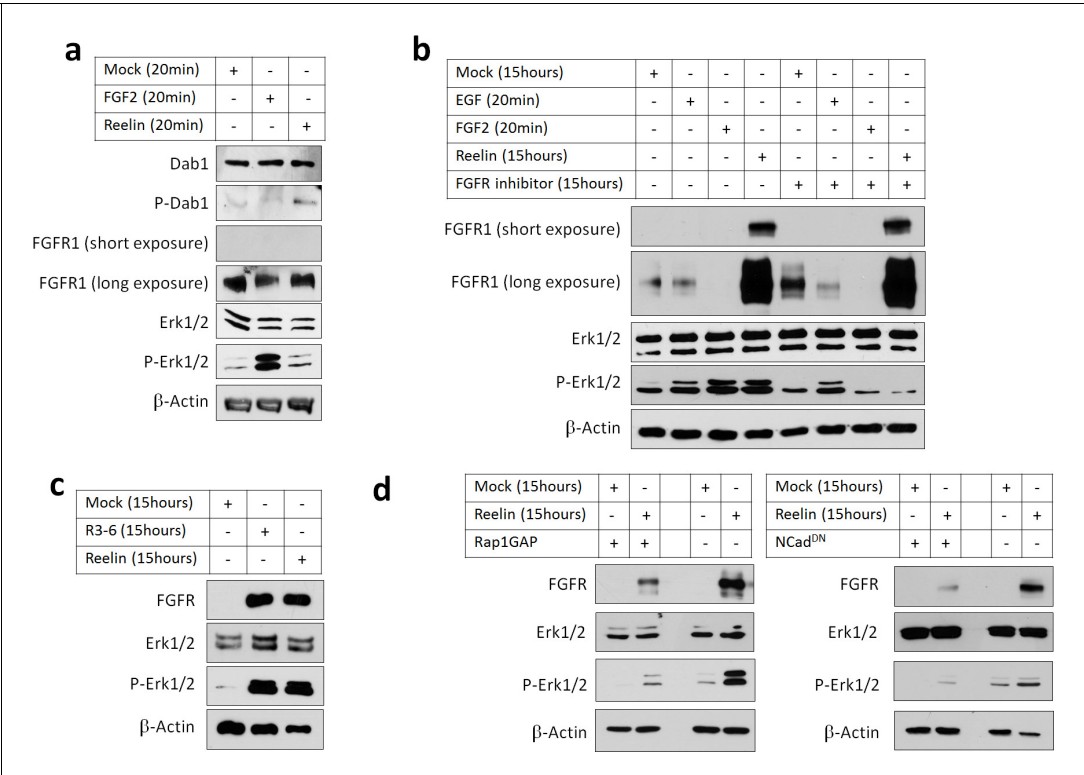

**Figure 8.** Reelin induces Rap1- and NCad-dependent FGFR and Erk1/2 activity in cortical neurons. E16.5 mouse cortical neurons were cultured for 3 days then stimulated with FGF2 or Reelin for different times. All experiments were repeated three times with similar results. (a) Short-term Reelin stimulation does not increase FGFR1 protein level or Erk1/2 phosphorylation. Neurons were stimulated for 20 min with 75 ng/ml FGF2, Mock- or Reelin-conditioned media. (b) Long-term Reelin stimulation increases FGFR1 protein level and FGFR1 and Erk1/2 phosphorylation dependent on FGFR1 kinase activity. Neurons were stimulated for 15 hr with Mock- or Reelin-conditioned media or for 20 min with FGF2 or EGF. FGFR inhibitor Debio1347 was used at a concentration of 5 μM for a total of 17 hr before cell lysis. (c) Reelin fragment R3-6 induces FGFR1 accumulation and Erk1/2 activation. Neurons were stimulated for 15 hr with Mock, R3-6 or Reelin-conditioned medium. (d) Long-term Reelin stimulation of FGFR1 *protein l*evel and Erk phosphorylation requires Rap1 and NCad. Neurons were electroporated with pCAG-Rap1GAP, pCAG-NCad$^{DN}$, or vector, incubated for 2 days, then stimulated with Mock- or Reelin-conditioned media for 15 hr and analyzed by Western blotting.

DOI: https://doi.org/10.7554/eLife.47673.023

The following figure supplement is available for figure 8:

**Figure supplement 1.** FGFR inhibitor does not inhibit Dab1 phosphorylation.

DOI: https://doi.org/10.7554/eLife.47673.024

dominant-negative mutants of Erk1 or Erk2, each of which inhibits both family members (*Watts et al., 1998*; *Li et al., 1999*; *Zampieri et al., 2007*). Over-expression of either Erk1$^{DN}$ or Erk2$^{DN}$, but not wildtype Erk1 or Erk2, induced partial migration arrest of multipolar neurons (*Figure 9a*). Overall our results suggest that multipolar migration of projection neurons requires a Reelin-Rap1-NCad-FGFR-Erk1/2 pathway.

## Discussion

Although much is known about the role of FGFRs in apical neural stem cells during neurogenesis and regional patterning of the cortex, the functions and regulation of FGFRs in neuronal migration have not been elucidated. Our findings implicate FGFRs in a signaling pathway that regulates the orientation of projection neuron multipolar migration during the development of the cerebral cortex (see our working model *Figure 9b*). When FGFRs are inhibited, multipolar neurons accumulate in the multipolar migration zone. This includes a defect in the orientation of multipolar migration with fewer cells facing the CP, followed by a delay in the multipolar to radial morphology transition and a

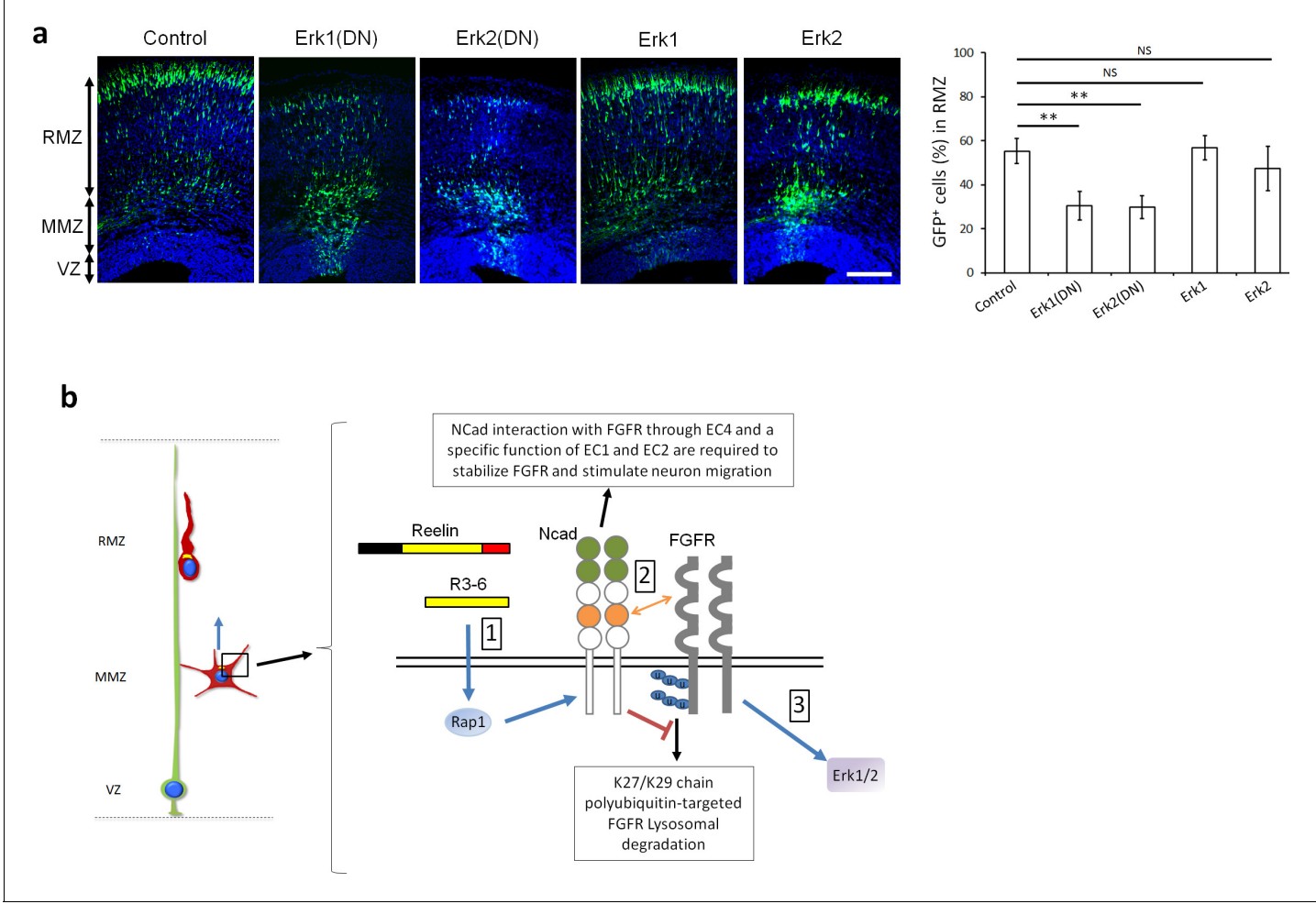

**Figure 9.** Erk1 and Erk2 regulate multipolar migration in vivo. (**a**) Erk1/2 inhibition impairs multipolar migration. E14.5 embryos were electroporated in utero with pCAG-GFP and dominant-negative (DN) or wildtype pNeuroD-Erk1/2. Cryosections were prepared three days later. The graph shows the percentage of cells in the RMZ (mean ± s.e.m.). *P* values: Erk1(DN): 1.4E-3, Erk2(DN): 1.2E-3, Erk1: 0.497, Erk2: 0.173. *n* = 4 Control, 6 Erk1(DN), 5 Erk2 (DN), 6 Erk1, 4 Erk2. Scale bar 100 μm; Error bars, s.e.m, **p<0.01, NS, not significant. (**b**) Working model. (1) Reelin, its central fragment R3-6, and possibly other signals activate Rap1 in multipolar neurons. Rap1 upregulates NCad on the cell surface. (2) The NCad fourth cadherin extracellular domain (EC4, orange color) binds FGFRs. This binding, together with a specific function of NCad EC1 and EC2 (green color) prevents FGFR polyubiquitination by mixed K27- and K29-linked polyubiquitin chains and lysosomal degradation. (3) Decreased FGFR ubiquitination causes FGFR accumulation and persistent activation of FGFR signaling pathways, including Erk1/2. Erk1/2 and maybe other effectors are required for the multipolar migration in vivo. The mechanisms by which NCad EC1-2 regulate FGFR stability and by which Erk1/2 regulate migration remain unknown. See text for discussion.

DOI: https://doi.org/10.7554/eLife.47673.025

The following source data is available for figure 9:

**Source data 1.** *Erk1/2 inhibition impairs multipolar migration.*

DOI: https://doi.org/10.7554/eLife.47673.026

consequent accumulation of cells at the MMZ. Previous studies indicated that the orientation of multipolar migration and the subsequent bipolar morphology transition is triggered by Reelin, or the R3-6 fragment, diffusing from the outer part of the cortex (*Jossin et al., 2004*; *Jossin et al., 2007*; *Uchida et al., 2009*; *Jossin and Cooper, 2011*). Reelin activates Rap1, and this in turn upregulates NCad on the neuron surface (*Jossin and Cooper, 2011*). However, the mechanism by which NCad stimulates the multipolar migration was unknown. One possibility was that NCad on migrating neurons engages in homophilic binding interactions with NCad on surrounding cells (*Kawauchi et al., 2010*; *Matsunaga et al., 2017*). However, our new evidence leads us to propose a model in which NCad-NCad *trans* interactions are not required. Instead, NCad binds in cis to FGFRs on the same

cell. This cell-autonomous binding inhibits FGFR K27/K29 polyubiquitination and lysosomal degradation, resulting in a large increase in FGFR abundance and prolonged activation of FGFR and Erk1/2 that is required for the multipolar migration. FGFR-dependent Erk1/2 stimulation may therefore be a key signal for orienting multipolar neurons towards the CP.

FGFRs have been implicated in other developmental cell movements, including migration of mesodermal and tracheal cells in *Drosophila* (*Gisselbrecht et al., 1996*; *Lebreton and Casanova, 2016*), neuroblasts in the mouse olfactory bulb (*Zhou et al., 2015*) and keratinocytes during repair of epidermal injury (*Meyer et al., 2012*). A possible role for FGFRs in cortical projection neuron migration has not been reported previously, perhaps because FGFRs are needed for early telencephalon patterning and neurogenesis, and because of functional redundancy (*Kang et al., 2009*; *Paek et al., 2009*). Cortical layering was defective in a mouse model expressing dominant-negative FGFR1 during neurogenesis but this could be secondary to the defective radial glia processes (*Shin et al., 2004*). By inhibiting FGFR signaling in postmitotic neurons, we avoided effects on neurogenesis or radial glia processes. Under these conditions, neurons were delayed in the multipolar migration zone, with randomized orientation, suggesting that FGFRs provide directional information. Conventionally, FGFRs are activated by FGFs and heparan sulfate proteoglycans (*Ornitz and Itoh, 2015*), so directional information could be provided by an FGF gradient, serving as a chemorepellent or attractant, as in the *Drosophila* trachea (*Lebreton and Casanova, 2016*). However, a large number of FGFs are expressed in the developing cortex (*Ford-Perriss et al., 2001*), making it challenging to identify which, if any, may be involved. It is also possible that NCad activates FGFRs in the absence of FGFs. Reelin could stimulate FGFR-dependent Erk1/2 activity in cultured neurons in the absence of added FGF, suggesting that external FGF may not be needed in vivo. The co-clustering of FGFRs with NCad may be sufficient to activate the receptor independently of FGF, with FGFRs acting as 'catalytic subunits' downstream of NCad. A similar mechanism has been suggested for activation by the co-receptor Klotho (*Lee et al., 2018*).

Our cell culture experiments suggest that NCad stabilizes cell-surface FGFR by inhibiting K27/K29-linked polyubiquitination and lysosomal degradation. Importantly, inhibiting K27-linked polyubiquitination by expression of a mutant ubiquitin rescued the migration of Rap1-inhibited neurons, suggesting that this mechanism also occurs in vivo. The involvement of K27 and K29 is unusual. Polyubiquitin chains assemble through any of seven lysine residues on the ubiquitin molecule, resulting in diverse outcomes for the target protein (*Fushman and Wilkinson, 2011*). While K11 and K48 polyubiquitination is linked to proteasomal degradation and K63 polyubiquitination to lysosomal degradation, the roles of K6-, K27-, K29-, and of K33-linked ubiquitin chains are less clear (*Sadowski et al., 2012*). K27 and K29 ubiquitin linkages have been implicated in protein interactions and protein degradation (*Chastagner et al., 2006*; *Ikeda and Kerppola, 2008*; *Zotti et al., 2011*; *Fei et al., 2013*; *Zhou et al., 2013*; *Birsa et al., 2014*; *Liu et al., 2014*). We found here that both K27 and K29 ubiquitin linkages are necessary for FGFR lysosomal degradation. This may involve mixed linear or mixed branched ubiquitin chains or single K27 and K29 chains attached to different lysine residues on the cytoplasmic tail of FGFRs (*Yau and Rape, 2016*). The E3 ligase and mechanism of lysosomal targeting remains unclear.

ECad also binds FGFRs but, unlike NCad, does not inhibit FGFR ubiquitination, protect FGFRs from degradation, or regulate multipolar migration in vivo. Using mutated receptors and chimeric proteins, we found that FGFRs bind both ECad and NCad and that binding is necessary but not sufficient to stabilize FGFRs and stimulate multipolar migration. Binding requires the fourth NCad EC domain, but FGFR stabilization and neuronal migration rescue also require the first two NCad domains. Their function is unclear. They may bind to a third partner in the complex that helps NCad inhibit FGFR degradation.

Our results indicated that Reelin does not activate ERK signaling directly but does so by stabilizing FGFR. ERK phosphorylation was delayed, presumably due to the time necessary to accumulate sufficient FGFR. A prolonged Erk1/2 activation might be required to induce a signal as is the case for EGF-induced neuronal differentiation of PC12 cells (*Traverse et al., 1994*). For instance, Erk1/2 signaling has been linked to transcription of matrix metalloproteinases genes (*Westermarck and Kahari, 1999*) and some metalloproteinases are essential for the organization of the cerebral cortex (*Jorissen et al., 2010*). Interestingly, Erk1 and Erk2 double mutant mice exhibit a defect in cortical lamination (*Imamura et al., 2010*). A cell-autonomous function of Erk1/2 signaling could not be

determined because of a failure in maintenance of the radial glia scaffolding. However, our results indicate that Erk1/2 activity is required in migrating neurons during the multipolar phase.

While we were writing this manuscript, FGFR2 was reported to regulate neuronal migration and spine density (*Szczurkowska et al., 2018*). Defective mice showed impaired core behaviors related to autism spectrum disorders. Interestingly, the Reelin pathway also regulates spine density and has been linked to autism (*Niu et al., 2008*; *Lammert and Howell, 2016*). Reelin was also reported to regulate the migration of dopaminergic neurons into the substantia nigra, with induction of bipolar morphology (*Vaswani et al., 2019*). Additional research will be needed to elucidate the mechanism of FGFRs in Reelin-induced migration and spine density and the link with autism spectrum disorders.

# Materials and methods

## Key resources table

| Reagent type (species) or resource | Designation | Source or reference | Identifiers | Additional information |
|---|---|---|---|---|
| Strain, strain background (*Escherichia coli*) | One shot TOP10 | Thermo Fisher Scientific | Cat #: C404010 | Chemically Competent Cells |
| Strain, strain background (*Mus musculus*) | CD1 | Charles River Laboratories | 022 | |
| Cell line (*Homo sapiens*) | HEK293T cells | ATCC | CRL-3216 | |
| Cell line (*Mus musculus*) | Embryonic primary mouse cortical neuron | This paper | N/A | Primary culture at E16,5. |
| Antibody | Anti-HA.11 clone 16B12 (Mouse monoclonal) | Eurogenetic | Cat# MMS-101R-500, RRID: AB_10063630 | WB (1:8000) IF (1:100) IP (1:400) |
| Antibody | Anti-Myc (Rabbit polyclonal) | Cell Signaling Technology | Cat# 2272, RRID: AB_10692100 | WB (1:5000) |
| Antibody | Anti-Myc-tag clone 9B11 (Mouse monoclonal) | Cell Signaling Technology | Cat# 2276, RRID: AB_331783 | IP (1:200) |
| Antibody | Anti mono- and polyubiquitinated conjugated, clone FK2 (Mouse monoclonal) | Enzo Life Science | Cat# BML-PW8810-0500, RRID: AB_2051891 | WB (1:1000) |
| Antibody | Anti-B-actin (Mouse monoclonal) | Thermo Fisher Scientific | Cat# MA5-15739, RRID: AB_10979409 | WB (1:5000) |
| Antibody | Anti- p44/42 MAPK (Erk1/2) (Rabbit polyclonal) | Cell Signaling Technology | Cat# 9102, RRID: AB_330744 | WB (1:5000) |
| Antibody | Anti-Phospho-p44/42 MAPK (Erk1/2) (Thr202/Tyr204) clone D13.14.4E (Rabbit monoclonal) | Cell Signaling Technology | Cat# 4370, RRID: AB_2315112 | WB (1:5000) |
| Antibody | Anti-FGFR1(D8E4) XP (Rabbit monoclonal) | Cell Signaling Technology | Cat# 9740, RRID: AB_11178519 | WB (1:500) |
| Antibody | Anti- Dab-1 (E1) (Mouse) | *Jossin et al., 2004* | N/A | WB (1:1000) |
| Antibody | Phospho-Tyrosine (P-Tyr-100) #9411(Mouse Monoclonal) | Cell Signaling Technology | Cat# 9411, RRID: AB_331228 | WB (1:1000) |
| Antibody | Anti-Reelin(G10) Mouse | *de Bergeyck et al., 1997* | N/A | WB (1:1000) |
| Antibody | Anti-DYKDDDDK (FLAG) Tag (FG4R) (Mouse monoclonal) | Thermo Fisher Scientific | Cat# MA5-15255, RRID: AB_2537646 | WB (1:5000) |

*Continued on next page*

*Continued*

| Reagent type (species) or resource | Designation | Source or reference | Identifiers | Additional information |
|---|---|---|---|---|
| Antibody | Anti-GFP (Rabbit polyclonal) | Thermo Fisher Scientific | Cat# A-11122, RRID: AB_221569 | WB (1:5000) |
| Antibody | Anti-Ki67 (Mouse monoclonal) | BD Biosciences | Cat# 556003, RRID: AB_396287 | IF (1:100) |
| Antibody | Anti- Sox2 (L1D6A2) (Mouse monoclonal) | Cell Signaling Technology | Cat# 4900, RRID: AB_10560516 | IF (1:100) |
| Antibody | Anti-Tbr2 (Rabbit polyclonal) | Abcam | Cat# ab23345, RRID: AB_778267 | IF (1:100) |
| Antibody | Anti-Satb2 (mouse | Abcam | Cat# ab51502 RRID: AB_882455 | IF (1:100) |
| Antibody | Anti-cleaved caspase3 (Rabbit polyclonal) | Cell Signaling Technology | Cat# 9661, RRID: AB_2341188 | IF (1:100) |
| Antibody | GM130 (Mouse monoclonal) | BD Biosciences | Cat# 610823, RRID: AB_398142 | IF (1:100) |
| Antibody | Anti-mouse IgG, HRP-linked Antibody (horse) | Cell Signaling Technology | Cat# 7076, RRID: AB_330924 | WB (1:5000) |
| Antibody | Anti-rabbit IgG, HRP-linked Antibody (goat polyclonal) | Cell Signaling Technology | Cat# 7074, RRID: AB_2099233 | WB (1:5000) |
| Antibody | Anti-Mouse IgG (H+L) Cross-Adsorbed Secondary Antibody, Alexa Fluor 488 (Goat Polyclonal) | invitrogen | Cat# A-11001, RRID: AB_2534069 | IF (1:100) |
| Antibody | Anti-Rabbit IgG (H+L) Antibody, Alexa Fluor 488 Conjugated (Goat polyclonal) | invitrogen | Cat# A-11008, RRID: AB_143165 | IF (1:100) |
| Antibody | Anti-Mouse IgG (H+L) Antibody, Alexa Fluor 568 Conjugated (goat polyclonal) | invitrogen | Cat# A-11004, RRID: AB_2534072 | IF (1:100) |
| Antibody | Anti-Rabbit IgG (H+L) Cross-Adsorbed Secondary Antibody, Alexa Fluor 568 (Goat polyclonal) | invitrogen | Cat# A-11011, RRID: AB_143157 | IF (1:100) |
| Antibody | Anti-Mouse IgG (H+L) Cross-Adsorbed Secondary Antibody, Alexa Fluor 647 (Goat polyclonal) | invitrogen | Cat# A-21235, RRID: AB_2535804 | IF (1:100) |
| Antibody | Anti-Rabbit IgG (H+L) Cross-Adsorbed Secondary Antibody, Alexa Fluor 647 (Goat polyclonal) | invitrogen | Cat# A-21244, RRID: AB_2535812 | IF (1:100) |
| Recombinant DNA reagent | NeuroD: Rap1GAP | *Jossin and Cooper, 2011* | N/A | |
| Recombinant DNA reagent | NeuroD: NCad CAG: NCad | *Jossin and Cooper, 2011* | N/A | |
| Recombinant DNA reagent | pBS mFgfr1 (CT#92) | Addgene | RRID: Addgene_14005 | mFgfr1 subcloned into pCAG and pNeuroD vectors. |
| Recombinant DNA reagent | CAG:FGFR2 NeuroD:FGFR2 | This paper | N/A | |
| Recombinant DNA reagent | CAG:FGFR3 NeuroD-FGFR3 | This paper | N/A | |

*Continued on next page*

*Continued*

| Reagent type (species) or resource | Designation | Source or reference | Identifiers | Additional information |
|---|---|---|---|---|
| Recombinant DNA reagent | CAG:FGFR1-DN NeuroD:FGFR1-DN | This paper | N/A | Deletion of the ICD, replace by GFP |
| Recombinant DNA reagent | CAG:FGFR2-DN NeuroD:FGFR2-DN | This paper | N/A | Deletion of the ICD, replace by GFP |
| Recombinant DNA reagent | CAG:FGFR3-DN NeuroD:FGFR3-DN | This paper | N/A | Deletion of the ICD, replace by GFP |
| Recombinant DNA reagent | pRK5:HA-ubiquitin-KO | Addgene | RRID: Addgene_17603 | |
| Recombinant DNA reagent | pRK5:HA-ubiquitin-K27 | Addgene | RRID: Addgene_22902 | |
| Recombinant DNA reagent | pRK5:HA-ubiquitin-K29 | Addgene | RRID: Addgene_22903 | |
| Recombinant DNA reagent | pHRSIN:6HIS-Ubi$^{WT}$-GFP | *Schaller et al., 2014* | N/A | |
| Recombinant DNA reagent | pHRSIN:6HIS-Ubi$^{K6R}$-GFP | *Schaller et al., 2014* | N/A | |
| Recombinant DNA reagent | pHRSIN:6HIS-Ubi$^{K11R}$-GFP | *Schaller et al., 2014* | N/A | |
| Recombinant DNA reagent | pHRSIN:6HIS-Ubi$^{K27R}$-GFP | *Schaller et al., 2014* | N/A | |
| Recombinant DNA reagent | pHRSIN:6HIS-Ubi$^{K29R}$-GFP | *Schaller et al., 2014* | N/A | |
| Recombinant DNA reagent | pHRSIN:6HIS-Ubi$^{K33R}$-GFP | *Schaller et al., 2014* | N/A | |
| Recombinant DNA reagent | pHRSIN:6HIS-Ubi$^{K48R}$-GFP | *Schaller et al., 2014* | N/A | |
| Recombinant DNA reagent | pHRSIN:6HIS-Ubi$^{K63R}$-GFP | *Schaller et al., 2014* | N/A | |
| Recombinant DNA reagent | pFLAG-CMV-hErk1 | Addgene | RRID: Addgene_49328 | hErk1 subcloned into pCAG and pNeuroD vectors |
| Recombinant DNA reagent | pFLAG-CMV-hErk1$^{K71R}$ | Addgene | RRID: Addgene_49329 | hErk1$^{K71R}$ subcloned into pCAG and pNeuroD vectors |
| Recombinant DNA reagent | p3XFLAG-CMV7-Erk2 | Addgene | RRID: Addgene_39223 | Erk2 subcloned into pCAG and pNeuroD vectors |
| Recombinant DNA reagent | p3XFLAG-CMV7-Erk2_KR | Addgene | RRID: Addgene_39224 | Erk2_KR subcloned into pCAG and pNeuroD vectors |
| Recombinant DNA reagent | NeuroD:NCad$^{\Delta EC4}$ CAG: NCad$^{\Delta EC4}$ | This paper | N/A | Deletion of residues 515–604 |
| Recombinant DNA reagent | CAG:NCad$^{DN}$ NeuroD:NCad$^{DN}$ | *Jossin and Cooper, 2011* | N/A | Deletion of residues 99–708 |
| Recombinant DNA reagent | CAG: Ncad$^{W161A}$ NeuroD: Ncad$^{W161A}$ | This paper | N/A | Codon 161 TGG replaced by GCG |
| Recombinant DNA reagent | CAG:Ecad NeuroD:ECad | This paper | N/A | |
| Recombinant DNA reagent | CAG:ENCad NeuroD: ENCad | This paper | N/A | |
| Recombinant DNA reagent | CAG:NECad NeuroD: NECad | This paper | N/A | |
| Recombinant DNA reagent | CAG: NENCad NeuroD: NENCad | This paper | N/A | |

*Continued on next page*

*Continued*

| Reagent type (species) or resource | Designation | Source or reference | Identifiers | Additional information |
|---|---|---|---|---|
| Recombinant DNA reagent | CAG: ENNcad NeuroD: ENNcad | This paper | N/A | |
| Recombinant DNA reagent | pLKO.1.shFGFR1 | Sigma | Cat #: TCRN0000378435 | Target sequence: 5'-CTGGCTGGAGTC TCCGAATAT-3' |
| Recombinant DNA reagent | pLKO.1.shFGFR2 | Sigma | Cat #: TRCN0000023715 | Target sequence: 5'-GCCAGGGATATC AACAACATA-3' |
| Recombinant DNA reagent | pLKO.1.shFGFR3 | Sigma | Cat #: TRCN0000363373 | Target sequence: 5'CCACTTCAGTGT GCGTGTAAC-3' |
| Recombinant DNA reagent | pCMV:Reelin and pCMV:R3-6 | *Jossin et al., 2004* | N/A | |
| Peptide, recombinant protein | hFGF-2 | PeproTech | Cat #: 100-18B | |
| Peptide, recombinant protein | mEGF-2 | PeproTech | Cat #: 315–09 | |
| Chemical compound, drug | Protease inhibitor cocktail | Roche | Cat# 05056489001 | |
| Chemical compound, drug | Phosphatase inhibitor cocktail | Roche | Cat #: A32957 | |
| Chemical compound, drug | B27 | invitrogen | Cat #: 17504–044 | |
| Chemical compound, drug | Epoxomycin | Sigma | Cat #: E3652 | |
| Chemical compound, drug | Leupeptin | Carl Roth | Cat #: CN33 | |
| Chemical compound, drug | Debio1347 | Selleckchem | Cat #: S7665 | |
| Chemical compound, drug | Penicillin-streptomycin | Gibco | Cat #: 11548876 | |
| Commercial assay or kit | Plasmid DNA Purification Mini Prep Kit | Intron Biotechnology | Cat# 17098 | |
| Commercial assay or kit | Quick Gel extraction Kit | Thermo Fisher Scientific | Cat# K2100-12 | |
| Commercial assay or kit | HiPure Plasmid Maxiprep Kit | Thermo Fisher Scientific | Cat# K2100-07 | |
| Software, algorithm | Image J | NIH | N/A | |
| Other | DAPI staining | Sigma | Cat #: D9542 | |
| Other | PolyJet In Vitro DNA Transfection Reagent | Signagen | Cat #: SL100688 | |
| Other | Dynabeads protein A | invitrogen | Cat #: 10001D | |
| Other | Dynabeads protein G | İnvitrogen | Cat #: 1003D | |
| Other | Super signal West Pico PLUS chemuluminescent substrate | Thermo Scientific | Cat #: 34578 | |
| Other | O.C.T. | VWR | Cat # 361603E | |
| Other | DMEM-F-12 | Gibco | Cat #: 21331–020 | |

*Continued on next page*

*Continued*

| Reagent type (species) or resource | Designation | Source or reference | Identifiers | Additional information |
|---|---|---|---|---|
| Other | DMEM, high glucose | Gibco | Cat #: 41965–039 | |
| Other | CL-X Posure film | Thermo fisher | Cat #: 34091 | |

## Mice

CD1 mice were bred in standard conditions and animal procedures were carried out in accordance with European guidelines and approved by the animal ethics committee of the Université catholique de Louvain.

## In utero electroporation

In utero microinjection and electroporation was performed at E14.5 essentially as described (*Tabata and Nakajima, 2001*), using timed pregnant CD-1 mice. Timed-pregnant mice were anesthetized and each uterus was exposed under sterile conditions. Plasmid solutions containing 1 µg/µl of DNA were injected into the lateral ventricles of the embryos using a heat-pulled capillary. Needles for injection were pulled from Wiretrol II glass capillaries (Drummond Scientific) and calibrated for 1 µl injections. DNA solutions were mixed in 10 mM Tris, pH 8.0, with 0.01% Fast Green. Forceps-type electrodes (Nepagene) with 5 mm pads were used for electroporation (five 50 ms pulses of 45 V using the ECM830 electroporation system, Harvard Apparatus). Embryos were placed back into the abdominal cavity, and mice were sutured.

## Histology and immunofluorescence

Embryos were collected at E16.5 or E17.5. Brains were dissected and successful electroporations were chosen by epifluorescence microscopy. Positive brains were fixed in a 3.7% paraformaldehyde (PFA) in phosphate-buffered saline (PBS) solution and cryoprotected in a 30% sucrose/PBS solution overnight at 4°C. Brains were frozen in optimal cutting temperature compound (OCT), and sectioned with a cryostat at 14-µm-thickness. Sections were placed on slides, permeabilized for 30 min in 0.4% Triton X-100/PBS then blocked for 30 min with 5% normal goat serum (NGS) in 0.4% Triton X100/PBS. Primary antibodies were diluted in 0.4% Triton X100/PBS incubated on slides overnight at 4°C. Sections were washed 3 times for 5 min in 0.4% Triton X100/PBS. Secondary antibodies were diluted in 0.4% Triton X100/PBS and incubated for 1 hr at room temperature. Nuclei were stained with 4,6-diamidino-2-phenylindole (DAPI). Slides were washed three times as before and coverslipped with Fluoroshield with 1, 4-Diazabicyclo [2.2.2] octane (Sigma) as an anti-fade reagent. Images were acquired with an Olympus FV1000 confocal microscope.

## Isolation, culture and nucleofection of primary Cortical Neuron

Neurons were dissected from E16.5 mouse embryo telencephalons. Cells were plated in DMEM-F12 medium (Fisher) with 2% B27 supplement (Fisher) and Penicillin-Streptomycin (Fisher) on 12 well-plate coated with poly-D-lysine (Sigma) and E-C-L (entactin-collagen IV-laminin) Cell Attachment Matrix (Upstate Biotechnology) at a density of $2 \times 10^6$ cells per dish. Cultures were maintained at 37°C in a 5% $CO_2$ incubator. After 2 days in culture, neurons were stimulated with partly-purified Reelin or Mock-conditioned media (see below), EGF (Pepro Tech, 100 ng/mL), FGF-2 (Pepro Tech, 75 ng/mL). Debio1347 (5 µM, Selleckchem) was used to inhibit FGFR1, 2 and 3 (*Nakanishi et al., 2014*; *Nakanishi et al., 2015*). Cells were lysed with ice-cold NP-40 buffer (150 mM NaCl, 50 mM Tris-HCl (pH 8.0), 1% NP-40, 5 mM EDTA) supplemented with protease and phosphatase inhibitor cocktail (Roche). For plasmid DNA transfection, cells were Amaxa nucleoporated with 5 µg of plasmid DNA in 0.2 mm cuvette using A-033 program and 100 µl of electroporation buffer containing 120 mM $Na_2HPO_4/NaH_2PO_4$ pH 7.2, 5 mM NaCl, 5 mM KCl, 20 mM $MgCl_2$, and 0.5 mM reduced glutathione.

## Production of recombinant reelin and R3-6

HEK293T cells cultured in Dulbecco modified Eagle medium (Fisher) with 10% fetal bovine serum (Fisher) were transfected with the Reelin or R3-6 cDNA constructs (*Jossin et al., 2004*), using Polyjet

(Tebu-Bio). After 24 hr, the medium was replaced with a serum-free medium, which was collected 2 days later and stored at 4°C in the presence of a protease inhibitor cocktail (Complete, Roche). Prior to use, the supernatants were concentrated using Amicon Ultra columns with 100,000-molecular weight cutoff filters (Millipore) to reach the approximate concentration of 400 pM, which was estimated as described previously (*Jossin et al., 2004*), and dialyzed against culture medium by drop dialysis (Millipore VSWP02500). Mock solutions were prepared from control transfected HEK293T cells and used to control for potential co-purifying proteins.

## Immunoprecipitation and western blot

Transfected 293T cells were lysed with ice-cold NP-40 buffer supplemented with protease and phosphatase inhibitor cocktail (Roche). Lysates were clarified by centrifugation at 14,000Xg for 10 min at 4°C. Antibodies were added to the lysates for 2 hr at 4°C. Dynabeads protein A or protein G magnetic beads (Invitrogen) were washed three times in PBS then blocked in 1% BSA/PBS for 2 hr at 4°C. Beads were washed twice with PBS and once in NP-40 buffer then added into cell lysate mixture and incubated overnight at 4°C. Beads were washed three times with NP-40 lysis buffer. Proteins were eluted by boiling for 5 min in polyacrylamide gel electrophoresis loading buffer and analyzed by sodium dodecyl sulfate (SDS)-polyacrylamide gel electrophoresis.

Proteins were separated by SDS-gel electrophoresis then transferred to nitrocellulose membrane (Amersham Biosciences) by electroblotting. Membranes were blocked in 5% skimmed milk and 0.05% Tween 20 in PBS for 1 hr and incubated overnight at 4°C with antibodies. After three washing steps in PBS with 0.05% Tween 20, membranes were incubated with horseradish peroxidase-conjugated secondary antibodies (DAKO) in blocking solution for 1 hr at room temperature and washed three times. Membranes were treated with the SuperSignal West Pico chemiluminescent substrate (Pierce) and exposed to Hyperfilm ECL (Amersham Biosciences).

## Antibodies

The following antibodies were used for immunofluorescence, immunoprecipitation or biochemistry: mouse anti-HA.11 clone 16B12 monoclonal antibody (Eurogentec), rabbit anti-myc (Cell Signaling), anti mono- and poly-ubiquitinated antibody clone FK2 (Enzo), mouse anti-β-Actin(Thermo Pierce), rabbit anti-Erk1/2 (Cell Signaling), rabbit anti p44/42 Erk (Thr 202/Tyr 204) monoclonal antibody (Cell Signaling), rabbit anti-FGFR1(D8E4) monoclonal antibody (Cell Signaling), rabbit anti-FGFR (phosphor-Tyr653/654) (Cell Signaling), mouse anti-Dab1 (E1) (*Jossin et al., 2004*), mouse anti-phospho-tyrosine antibody (Cell Signaling), mouse anti-Reelin (G10) (*de Bergeyck et al., 1997*), mouse anti-Flag (Thermo Pierce), rabbit anti-GFP (Invitrogen), mouse anti-Ki67 (Beckton Dickinson), mouse anti-Sox2 (Cell Signaling), Rabbit anti Tbr2 (Abcam), mouse anti-Satb2 (Abcam), rabbit anti-cleaved caspase 3 (Cell Signaling), mouse GM130 (Beckton Dickinson).

Goat secondary antibodies labeled with Alexa 488, 568, and 647 (Invitrogen) for immunofluorescence. Goat anti-mouse or anti-rabbit horseradish peroxidase-conjugated secondary antibodies (Cell Signaling) for biochemistry.

## Vector constructions

Rap1GAP and NCad sequences inserted into the pNeuroD vector were described previously (*Jossin and Cooper, 2011*). Plasmid containing the coding sequences for FGFR1 (Addgene plasmid # 14005) was used as template to insert the sequences into pCAG or pNeuroD vectors. FGFR2 and FGFR3 were amplified from E16.5 embryonic mouse cortex. Dominant negative forms of FGFR1, 2 and 3 contain the transmembrane and extracellular domains while the intracellular domains where replace by GFP. pRK5-HA-Ubiquitin-KO was a gift from Ted Dawson (Addgene plasmid # 17603). pRK5-HA-Ubiquitin-K27 and pRK5-HA-Ubiquitin-K29 were a gift from Sandra Weller (Addgene plasmid # 22902 and # 22903). Wild-type and all single lysine to arginine ubiquitin mutant vectors pHRSIN-6HIS-Ubi$^{WT}$-GFP, Ubi$^{K6R}$-GFP, Ubi$^{K11R}$-GFP, Ubi$^{K27R}$-GFP, Ubi$^{K29R}$-GFP, Ubi$^{K33R}$-GFP, Ubi$^{K48R}$-GFP, Ubi$^{K63R}$-GFP were provided by R Rezsohazy with the permission of M Malim (*Schaller et al., 2014*). pFLAG-CMV-hErk1, pFLAG-CMV-hErk1$^{K71R}$, p3xFlag-CMV7-Erk2, p3xFlag-CMV7-Erk2_KR were a gift from Melanie Cobb (Addgene plasmid # 49328, # 49329, # 39223, # 39224) and subcloned into pCAG and pNeuroD vectors. NCad$^{\Delta EC4}$ was made by PCR using junction primers at the KpnI site already present in the sequence resulting in the deletion of residues 515–

604. NCad$^{W161A}$ was made by site-directed mutagenesis with oligo 5'-GCT CTA CAA AGG CAG AAG CGA GAC GCG GTC ATC CCG CCA ATC AAC-3' and its reverse complement, changing codon 161 from TCG to GCG and introducing a silent mutation to destroy an EarI restriction site that was used for screening. NCad$^{DN}$ contains the transmembrane and intracellular domains (deletion of residues 99–708) and was described before (*Jossin and Cooper, 2011*). ENCad and NECad were made by using PCR using junction primers that inserted SpeI sites five residues N-terminal to the transmembrane domain (between ECad codons 704 and 705, and between NCad codons 717 and 720). NENCad and ENNCad were made using overlap extension PCR with recombination junctions between residues 420–421 of NCad and 415–416 of ECad. All cadherin constructs were of murine origin, terminated with an HA tag and cloned into pCAG and pNeuroD vectors. Codon numbers are given from the initiator methionine. Several FGFR targeting shRNAs (pLKO.1 plasmids either from Sigma or generously provided by Slobodan Beronja and shRNA expression vectors kindly provided by Laura Cancedda and Giovanni Piccoli; *Szczurkowska et al., 2018*) were tested and the most efficient shRNAs were selected for in vivo experiments. The most effective shRNAs are shFGFR1: clone ID TRCN0000378435 (target sequence 5'-CTGGCTGGAGTCTCCGAATAT-3'), shFGFR2: clone ID TRCN0000023715 (target sequence 5'-GCCAGGGATATCAACAACATA-3') and shFGFR3: clone ID TRCN0000363373 (target sequence 5'-CCACTTCAGTGTGCGTGTAAC-3').

## Cell line culture

HEK293T cells (ATCC) maintained in Dulbecco's modified Eagle's medium (DMEM) supplemented with 10% FBS, 100 IU/mL penicillin, and 100 µg/mL streptomycin were transfected using Polyjet (Tebu-bio). Cells were cultured at 37°C under 5% CO2, and are mycoplasma-free. The following inhibitors were used: Epoxomycin, (Sigma), Leupeptin (Roth), Debio1347 (Selleckchem).

## Statistical analysis

Statistical analysis made use of Student's t-test across N samples, where N is the number of embryos or experiments as defined in the figure legends.

## Acknowledgements

We thank Sasha Strait for excellent technical assistance. YJ is a Fonds National de la Recherche Scientifique (FNRS) investigator. EK, ECJ and AC are supported by Fonds pour la Formation à la Recherche dans l'Industrie et dans l'Agriculture (FRIA) fellowships. This work was supported by grants J.0129.15, J.0179.16 and T.0243.18 from the FNRS and R01-NS080194 and GM109463 from the National Institutes of Health.

## Additional information

### Competing interests

Jonathan A Cooper: Senior editor, *eLife*. The other authors declare that no competing interests exist.

### Funding

| Funder | Grant reference number | Author |
|---|---|---|
| Fonds De La Recherche Scientifique - FNRS | J.0129.15 | Yves Jossin |
| Fonds De La Recherche Scientifique - FNRS | J.0179.16 | Yves Jossin |
| Fonds De La Recherche Scientifique - FNRS | T.0243.18 | Yves Jossin |
| Fonds pour la Formation à la Recherche dans l'Industrie et dans l'Agriculture | | Elif Kon Elisa Calvo-Jimenez Alexia Cossard |
| National Institutes of Health | R01-NS080194 | Jonathan A Cooper |

| National Institutes of Health | GM109463 | Jonathan A Cooper |

The funders had no role in study design, data collection and interpretation, or the decision to submit the work for publication.

## Author contributions

Elif Kon, Data curation, Formal analysis, Validation, Investigation, Visualization, Performed most of the biochemistry and histology experiments, Assisted with preparing the manuscript; Elisa Calvo-Jiménez, Data curation, Formal analysis, Validation, Investigation, Visualization, Assisted with the biochemistry and histology experiments, Performed the primary cortical neuron nucleofection experiments, Assisted with preparing the manuscript; Alexia Cossard, Investigation, Visualization, Assisted with the biochemistry and histology experiments, Assisted with preparing the manuscript; Youn Na, Investigation, Assisted with the biochemistry and histology experiments, Assisted with preparing the manuscript; Jonathan A Cooper, Writing—review and editing, Reviewed and edited the final version of the manuscript, Assisted with preparing the manuscript; Yves Jossin, Conceptualization, Resources, Data curation, Formal analysis, Supervision, Funding acquisition, Validation, Investigation, Visualization, Methodology, Writing—original draft, Project administration, Writing—review and editing, Conceived and supervised the project, Designed all experiments, Wrote the original draft of the manuscript, Reviewed and edited the final version of the manuscript, Performed the in vivo experiments

## Author ORCIDs

Jonathan A Cooper https://orcid.org/0000-0002-8626-7827
Yves Jossin https://orcid.org/0000-0001-8466-7432

## Ethics

Animal experimentation: CD1 mice were bred in standard conditions and animal procedures were carried out in accordance with European guidelines and approved by the animal ethics committee of the Université Catholique de Louvain under the protocol number: 2017/UCL/MD/009.

## Decision letter and Author response

Decision letter https://doi.org/10.7554/eLife.47673.029
Author response https://doi.org/10.7554/eLife.47673.030

# Additional files

## Supplementary files
• Transparent reporting form
DOI: https://doi.org/10.7554/eLife.47673.027

## Data availability
All data generated or analyzed during this study are included in the manuscript and supporting files.

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
