## [Decision Letter]

[Editors’ note: a previous version of this study was rejected after peer review, but the authors submitted for reconsideration. The first decision letter after peer review is shown below.]

Thank you for submitting your work entitled "FGFR ubiquitination and degradation controls neuronal migration in vivo" for consideration by *eLife*. Your article has been reviewed by two peer reviewers, and the evaluation has been overseen by Ivan Dikic as the Reviewing and Senior Editor. The following individual involved in review of your submission have agreed to reveal their identity: Mary E Hatten (Reviewer #2).

Our decision has been reached after consultation between the reviewers. Based on these discussions and the individual reviews below, we regret to inform you that your work will not be considered further for publication in *eLife*.

Summary:

The manuscript studies the role of FGF receptors in the migration of cortical projection neurons. The authors show that interaction of FGFR with N-Cadherin, CDH2, seems to protect FGFR from ubiquitination and subsequent degradation by lysosome. This leads to sustained FGFR levels and increased Erk1/2 downstream signaling which the authors propose regulates neuronal migration. Reelin seems important for FGFR-mediated Erk1/2 signaling, presumably by stabilizing and/or recruiting N-Cadherin to the plasma membrane.

While a number of interesting observations are presented, none directly assays FGFR function in cortical neuron migration. Rather, the data appear to show that disruption of FGFR and/or FGFR signaling stalls neuronal development at the multipolar stage, thereby preventing further differentiation and/or migration to form the neuronal layers. The experimental design and performance were nice, but the experiments are carried out mostly in HEK293 cells, not on developing cortical neurons. In addition, from the currently presented data it is difficult to see a coherent model for how the FGFRs actually function in locomotion (or a polarity defect at the multipolar stage). In total, the authors failed to demonstrate a direct function for FGFRs in migration in neurons.

The essential revisions to improve this work include a number of issues indicated in the reviews. In particular we feel that providing live-imaging data to resolve whether and to which extent FGFR signaling is required for migration versus polarization/differentiation is needed. In addition the authors need to repeat the biochemistry/pathways experiments in primary cortical cells and also draw a more coherent model of their functional relevance in vivo. Taken together, substantial amount of additional data is required in order to reach the high standard of *eLife*. If the authors wish to submit the work again to *eLife* following their revisions fulfilling the raised points at a later time the manuscript will be considered as a new submission.

Reviewer #1:

Kon and colleagues address important questions in cortical development and convincingly demonstrate an essential function for FGFR signaling in radial projection neuron migration. By using in utero electroporation (for gene knockdown) and structure function analysis, the authors elucidate mechanistic details. They show that interaction of FGFR with N-Cadherin, CDH2, seems to protect FGFR from ubiquitination and subsequent degradation by lysosome. This leads to sustained FGFR levels and increased Erk1/2 downstream signaling which regulates neuronal migration. Interestingly, Reelin is an important upstream regulatory component and seems important for FGFR-mediated Erk1/2 signaling, presumably by stabilizing and/or recruiting N-Cadherin to the plasma membrane.

Overall the manuscript by Kon et al. investigates timely aspects in cortical development. The manuscript is well written and the findings illustrated nicely. Some of the claims could be further substantiated by addressing the following points:

1) The authors propose that FGFR signaling is important for polarization and/or orientation of neurons in the MMZ. However the actual data supporting this claim directly is a bit slim. In Figure 1B the authors stain and assay for the location of Golgi but it is very difficult to appreciate the defect in orientation in relation to the cortical plate. Perhaps more important would be to document and quantify the disoriented direction of multipolar migration by life-imaging. This piece of data will be critical since it will provide mechanistic insight at the cellular level and further contribute to the conceptual advance of the study.

2) Figure 5B could be improved by showing the real microscopic images that relate to the quantification chart.

3) The Discussion ends a bit abruptly and a conclusive paragraph is needed. The authors should also discuss their findings in a somewhat broader context. They should elaborate how their findings extend and complement the current model of FGFR function in neuronal migration in vivo, which was recently published by the Cancedda lab.

Reviewer #2:

The manuscript by Cooper and colleagues "FGFR ubiquitination and degradation controls neuronal migration in vivo" presents a series of experiments on cortical development the stated purpose of which is to investigate the role of FGF receptors in the migration of cortical projection neurons. While a number of interesting observations are presented, none directly assays FGFR function in cortical neuron migration and the various pathways investigated do not inform a coherent model of how FGFRs would directly function in neuronal locomotion. Rather, the data appear to show that disruption of FGFR function stalls neuronal development at the multipolar stage, thereby preventing further differentiation and/or migration to form the neuronal layers. Also, while the pathways that impact FGFR dependent ERK1/2 phosphorylation and degradation are interesting vis a vis downstream signaling pathways, and the experiments are nicely done, the experiments are carried out in HEK293 cells, not on developing cortical neurons. The paper fails to demonstrate a direct function for FGFRs in migration.

Specific issues include the following:

1) Expression of a dominant negative form of the receptor is not as clean as a conditional genetic experiment in cortical projection neurons. As is true with many electroporation approaches, the identity of the neuronal populations that are affected are not specified.

2) There is no direct evidence that the FGFR family members act redundantly during the transition from the multipolar to bipolar stage and migration into the cortical plate. Can the dominant negative FGFR dimerize with receptors other than FGFR family members or N-Cad? Such interactions could contribute to the phenotypes observed.

3) Does NCad-dEC4 bind to receptors other than FGFRs?

4) Assays to identify binding partners, e.g. IP/mass spec experiments using lysates from primary cortical neurons expressing the dominant negative FGFR and NCad-dEC4 proteins, would be more informative.

5) While the HEK293T cells are a great starting point for the biochemistry, it is critical to perform the protein interaction and expression assays using purified cortical neurons or at least cultured primary cortical neurons. The fact that these changes, or lack of changes in some cases, occur in HEK293T cells does not guarantee that this is the case in post-mitotic cortical neurons.

6) There appears to be a disconnect between the Rap1/CDH2/FGFR and Reelin/FGFR/ERK1 and 2 signaling pathways in this manuscript that are bridged by the following sentence and the authors' previous paper "The Rap1-dependent upregulation of NCad is triggered by Reelin, an extracellular ligand present in the MMZ (Jossin and Cooper, 2011)". This needs to be more clearly stated.

7) Why is there no control image for Figure 1B to show the Golgi orientation of control cells?

8) Figure 2A and 3D: Why is FGFR expression not visible in all lanes on the WB when the protein is overexpressed? If it is a matter of exposure time, it would be preferable to use longer exposures in order to see bands in all lanes that indicate FGFR expression. It would also be helpful to see loading controls like in Figure 3D.

9) Figure 3A and 3B: In contrast to Figure 2C, there is no increase in *FGFR1*-Myc when co-expressed with NCad-HA. If the DNA concentration is adjusted as noted in the legend, it would still be expected that the NCad-HA and NCad-W161A-HA lanes show similar levels of *FGFR1*-Myc, which they do not. *FGFR1* appears even lower when co-expressed with NCad-HA. On the other hand, NCad-dEC4, which is proposed to not interact with *FGFR1*, seems to increase *FGFR1* expression. I would suggest to show blots with consistent data, or remove those claims in the text.

10) The authors state that NCad-dEC4 does not interact with *FGFR1* and is unable to rescue migration defects in Rap1GAP cells, which indicates that NCad interacts with *FGFR1* through its EC4 domain. However, the data in Figure 5 shows that NCad increases *FGFR1* expression and promotes neuronal migration through its EC1-2 domains. The same figure also shows that the EC4 domain-containing chimeric E/NCad does not increase FGFR or promote migration. This is confusing. If NCad does interact with *FGFR1* via EC4 and promotes migration via EC1-2, it would suggest that homophilic binding of NCad or heterophilic binding to other proteins, in addition to heterophilic NCad:FGFR binding, plays a role in migration. Nevertheless, the authors present data indicating that homophilic NCad interactions do not have a role migration. These data need clarification.

11) Were the ERK1/2 constructs expressed in cortical neurons selectively expressed in post-mitotic cells?

12) The Discussion ends rather abruptly.

[Editors’ note: what now follows is the decision letter after the authors submitted for further consideration.]

Thank you for sending your article entitled "FGFR ubiquitination and degradation control neuronal migration in vivo" for peer review at *eLife*. Your article is being evaluated by two peer reviewers, and the evaluation is being overseen by a Reviewing Editor and Marianne Bronner as the Senior Editor.

The reviewers are somewhat split regarding the revision of this manuscript. In particular, reviewer 2 raises important issues regarding the specificity of the FGFR loss-of-function experiments and well as conclusions drawn from the imaging. In particular, it would be important to include a more specific conditional knock-down and better imaging including live imaging.

Given the list of essential revisions, including new experiments, the editors and reviewers invite you to respond within the next two weeks with an action plan and timetable for the completion of the additional work. We plan to share your responses with the reviewers and then issue a binding recommendation. The full reviews are listed below.

Reviewer #1:

Kon and colleagues address important questions in cortical development and convincingly demonstrate an essential function for FGFR signaling in radial projection neuron migration. More specifically the authors show that FGFR-mediated signaling regulates multipolar migration and the switch to bipolar morphology. By using in utero electroporation (for gene knockdown) and structure function analysis, the authors elucidated mechanistic details. They show that interaction of FGFR with N-Cadherin, CDH2, seems to protect FGFR from ubiquitination and subsequent degradation by lysosome. This leads to sustained FGFR levels and increased Erk1/2 downstream signaling. Interestingly, Reelin is an important upstream regulatory component and seems important for FGFR-mediated Erk1/2 signaling, presumably by stabilizing and/or recruiting N-Cadherin to the plasma membrane. The study by Kon et al. investigates timely aspects in cortical development.

This version of the manuscript improved, is very well written and the findings illustrated nicely. The authors added new and important data which greatly support and strengthen the main conclusions of the manuscript. The authors made an effort to clarify the writing in general and now explicitly state that FGFR signaling controls multipolar migration and subsequent switch to bipolar morphology. Given the convincing data and additional experimental evidence the current data set supports the main conclusions of the manuscript very well, even without direct live-imaging data. Overall this study is likely to be of great interest to the broader neuroscience community.

Reviewer #2:

Kon et al. revised their manuscript ʺFGFR ubiquitination and degradation controls neuronal migration in vivoʺ to reflect their observation that cortical neurons appear to stall at the multipolar stage when FGFR signaling is perturbed by transgenic expression of DN FGFR. They have done a nice job of streamlining the biochemical and in utero electroporation epistasis experiments, and performing more of the biochemical experiments in cultured cortical neurons as requested, to work out the signaling mechanism involved. While the manuscript overall has improved, there are several issues that I feel do not make it suitable for publication in *eLife* at this time and may be more suitable for a more specialized journal.

First, I still worry that expression of a DN FGFR, as opposed to conditional genetic loss of function experiments, may have some off-target effects. In addition, the finding by Szczurkowska et al., 2018, that *FGFR2* regulates neuronal migration and spine density suggests that the FGFRs may not act redundantly.

Second, I do not agree that the terminology "multipolar migration" is justified without more detailed analysis of the cellular mechanism involved. Better high magnification/resolution images of cell morphology at this stage, combined with live imaging, would be required to determine why and how these cells are stalling.

---

## [Author Response]

[Editors’ note: the author responses to the first round of peer review follow.]

Summary:The manuscript studies the role of FGF receptors in the migration of cortical projection neurons. The authors show that interaction of FGFR with N-Cadherin, CDH2, seems to protect FGFR from ubiquitination and subsequent degradation by lysosome. This leads to sustained FGFR levels and increased Erk1/2 downstream signaling which the authors propose regulates neuronal migration. Reelin seems important for FGFR-mediated Erk1/2 signaling, presumably by stabilizing and/or recruiting N-Cadherin to the plasma membrane.While a number of interesting observations are presented, none directly assays FGFR function in cortical neuron migration. Rather, the data appear to show that disruption of FGFR and/or FGFR signaling stalls neuronal development at the multipolar stage, thereby preventing further differentiation and/or migration to form the neuronal layers. The experimental design and performance were nice, but the experiments are carried out mostly in HEK293 cells, not on developing cortical neurons. In addition, from the currently presented data it is difficult to see a coherent model for how the FGFRs actually function in locomotion (or a polarity defect at the multipolar stage). In total, the authors failed to demonstrate a direct function for FGFRs in migration in neurons.The essential revisions to improve this work include a number of issues indicated in the reviews. In particular we feel that providing live-imaging data to resolve whether and to which extent FGFR signaling is required for migration versus polarization/differentiation is needed. In addition the authors need to repeat the biochemistry/pathways experiments in primary cortical cells and also draw a more coherent model of their functional relevance in vivo. Taken together, substantial amount of additional data is required in order to reach the high standard of eLife. If the authors wish to submit the work again to eLife following their revisions fulfilling the raised points at a later time the manuscript will be considered as a new submission.

We thank the editor and reviewers for the summary. In response we have performed additional experiments and rewritten the text to improve logical flow. In summary:

1) We agree that disrupting FGFR signaling stalls or delays neurons at the multipolar stage. While our new data show that the phenotype is not due to a defect in differentiation, the evidence points to a failure before multipolar neurons become bipolar. This is the same phenotype as we described before when we inhibited Reelin receptors, Rap1 or N‐cadherin (Ncad) (Jossin and Cooper, 2011). In those experiments, we tracked neurons in slice cultures and found that migration speed was normal but the migration paths were more erratic, with more neurons moving down or sideways compared with controls. While we now lack the facilities to image neurons in slice culture, we show that blocking FGFR signaling has a similar effect as blocking Rap1 or NCad on neuron morphology and orientation near the top of the multipolar zone. Moreover, FGFR expression rescues multipolar neurons with inactive Rap1. Since Rap1 and NCad‐dependent FGFR activation by Reelin stimulates Erk, and since inhibiting Erk also delays neurons at the multipolar stage, we suspect that FGFRs may be needed for Erk‐dependent signals activated by Reelin in multipolar neurons. These signals might include expression of genes required for polarization, orientation or migration, but it would likely require single cell RNA‐Seq to identify a distinct differentiation state. To avoid misleading the reader as to the precise nature of the migration defect, we have now replaced “defect in neuronal migration” with “defect in multipolar migration”.

2) The original paper presented in vivo results, from in utero electroporation, showing:

a) Dominant‐negative FGFRs disturb the migration of neurons with multipolar morphology and impair their orientation towards the cortical plate.

b) Wildtype FGFR expression rescues migration of neurons with inhibited Rap1.

c) Ncad‐mediated cell‐cell adhesion is not needed for neuron migration in vivo.

d) NCad that cannot bind FGFRs does not rescue neurons with inhibited Rap1.

e) Over‐expressing ubiquitin K27R inhibits FGFR downregulation and rescues Rap1inhibited neurons.

f) While both ECad and NCad bind FGFRs, ECad does not prevent FGFR degradation and does not rescue multipolar migration in vivo unless it carries the NCad EC1‐2 region.

g) Dominant‐negative but not wildtype Erk1 or Erk2 inhibits multipolar neuron migration in vivo.

These in vivo experiments were complemented by experiments with cultured embryonic cortical neurons showing that long‐term stimulation with Reelin increases endogenous *FGFR1* expression and activates Erk, dependent on FGFR kinase activity, and that endogenous FGFRs are degraded by the lysosome pathway. Transfected HEK293 cells were used for biochemical experiments to validate the NCad mutants used, to confirm that NCad binds to and inhibits ubiquitination and degradation of FGFRs, and to identify the ubiquitin linkage in polyubiquitinated FGFR. In restructuring the paper, we have moved many of the HEK293 experiments to supplementary figures, to focus attention on the in vivo and neuron culture experiments. We also provide the following new data:

a) Neuron proliferation, apoptosis, and differentiation state, as determined with

*Sox2*, Tbr2 and Satb2, are unaffected when FGFRs are inhibited in vivo (new Figure 2A).

b) Inhibition of FGFRs in vivo affects the morphological switch from multipolar to bipolar (new Figure 2C).

c) Transfected FGFR levels decrease when Rap1 is inhibited in utero (new Figure 3B).

d) Endogenous FGFR levels increase when NCad is expressed in primary neurons (new Figure 3C).

e) Increased FGFR protein levels and activation of Erk in Reelin‐stimulated neurons requires Rap1 and NCad (new Figure 8D).

Other figures have been updated with improved data. We hope that the reviewers will understand that repeating all the biochemical experiments in neurons would be very time consuming and will excuse the remaining HEK293 experiments.

Reviewer #1:[…] Overall the manuscript by Kon et al. investigates timely aspects in cortical development. The manuscript is well written and the findings illustrated nicely. Some of the claims could be further substantiated by addressing the following points:1) The authors propose that FGFR signaling is important for polarization and/or orientation of neurons in the MMZ. However the actual data supporting this claim directly is a bit slim. In Figure 1B the authors stain and assay for the location of Golgi but it is very difficult to appreciate the defect in orientation in relation to the cortical plate. Perhaps more important would be to document and quantify the disoriented direction of multipolar migration by life-imaging. This piece of data will be critical since it will provide mechanistic insight at the cellular level and further contribute to the conceptual advance of the study.

We agree that live imaging would be a great addition to this study. Unfortunately, we no longer have the facilities for live imaging. Nevertheless, we improved the quality of the images provided in the new Figure 2B so the reader could better appreciate the phenotype. We went further in the description of the phenotype by showing a delay in the morphological transition from multipolar to bipolar cell (new Figure 2C). While the phenotype is very similar to that caused by inhibiting Rap1 or NCad (Jossin and Cooper, 2011),, we have bolstered the evidence that FGFR function is linked to the Reelin/Rap1/NCad pathway by adding new data: Reelin increases FGFR signaling in primary cortical neuron cultures, in an NCad‐dependent manner; Rap1 inhibition in vivo decreases FGFR protein level in multipolar neurons; NCad over‐expression in neurons increases FGFR protein level; FGFR inhibition does not alter cell fate, proliferation or survival in vivo.

2) Figure 5B could be improved by showing the real microscopic images that relate to the quantification chart.

We thank the reviewer for their comments. The requested pictures have been added (new Figure 6B).

3) The Discussion ends a bit abruptly and a conclusive paragraph is needed. The authors should also discuss their findings in a somewhat broader context. They should elaborate how their findings extend and complement the current model of FGFR function in neuronal migration in vivo, which was recently published by the Cancedda lab.

The interesting paper from Szczurkowska et al. shows that inhibiting FGFR expression impairs migration of late‐born neurons and decreases spine density after neurons mature. The first result is expected in light of our observation of a defect in multipolar migration. The second result also suggests parallels with Reelin signaling, because Reelin also increases spine density. Szczurkowska et al. have the added twist that another cell adhesion molecule, NEGR1 also inhibits FGFR degradation. This suggests that at least two pathways – Reelin/Rap1/NCad and NEGR1 – stabilize FGFRs and increase signaling. We have added a concluding paragraph commenting on this new paper.

Reviewer #2:The manuscript by Cooper and colleagues "FGFR ubiquitination and degradation controls neuronal migration in vivo" presents a series of experiments on cortical development the stated purpose of which is to investigate the role of FGF receptors in the migration of cortical projection neurons. While a number of interesting observations are presented, none directly assays FGFR function in cortical neuron migration and the various pathways investigated do not inform a coherent model of how FGFRs would directly function in neuronal locomotion. Rather, the data appear to show that disruption of FGFR function stalls neuronal development at the multipolar stage, thereby preventing further differentiation and/or migration to form the neuronal layers.

We never stated that FGFRs were involved in neuronal locomotion but may have implied it by our use of the generic term “neuronal migration”. We use now the more precise term “multipolar migration”. Cortical projection neurons go through two types of migration: multipolar migration and bipolar migration (or locomotion). Our results show that FGFRs are important for the multipolar migration that occurs before locomotion. The results provided here suggest that the Reelin/Rap1/Ncad pathway regulates FGFRs to control multipolar migration and our previous studies on Rap1‐inhibited neurons revealed no alteration in locomotion. We are sorry that it was confusing and we have made changes in the text to clarify this issue. Please see general point 1 above.

We have now added immunostaining results to show that neuronal differentiation, proliferation and survival are not affected in vivo (Figure 2A). However, we cannot exclude that the neurons are in a subtly different stage of differentiation that may be revealed by detailed gene expression analysis.

Also, while the pathways that impact FGFR dependent ERK1/2 phosphorylation and degradation are interesting vis a vis downstream signaling pathways, and the experiments are nicely done, the experiments are carried out in HEK293 cells, not on developing cortical neurons. The paper fails to demonstrate a direct function for FGFRs in migration.

In the previous version of the paper, we made use of cultured embryonic cortical neurons to show that long‐term stimulation with Reelin increases endogenous *FGFR1* expression and activates Erk, dependent on FGFR kinase activity, and that endogenous FGFRs undergo lysosomal, not proteasomal, degradation. In the re‐submission, we have added new data showing that NCad over‐expression in cortical neurons increases endogenous *FGFR1* protein level (new Figure 3C), that activation of Erk in Reelin simulated cortical neurons requires Rap1 and NCad (new Figure 8D), and that FGFR levels decrease when Rap1 is inhibited in utero (Figure 3B). HEK293 cells, which share features with neurons (Shaw et al. 2002, PMID 11967234), were used to validate the NCad mutants used, to confirm that NCad binds to and inhibits ubiquitination and degradation of FGFRs, and to identify the K27/29 ubiquitin linkage in polyubiquitinated FGFR. Such an extensive series of transfection experiments would be challenging in primary neurons. We feel that the preponderance of evidence supports the model depicted in Figure 9B, but there are gaps in our knowledge that we discuss frankly in the text.

Specific issues include the following:1) Expression of a dominant negative form of the receptor is not as clean as a conditional genetic experiment in cortical projection neurons.

It is true that the use of dominant‐negative forms has disadvantages but conditional knockout with post‐mitotic neuronal drivers such as Dcx‐Cre may not be effective if there is perdurance of FGFR mRNA or protein. In addition, *FGFR1*‐3 have overlapping activators and downstream effectors. By using the NeuroD promoter to inhibit the whole family of FGFRs we can rapidly inhibit FGFR signaling in migrating neurons without disrupting neurogenesis. There were no effects of electroporating the full-length FGFRs under similar conditions, suggesting that FGFR cytoplasmic domain functions are required for normal migration.

As is true with many electroporation approaches, the identity of the neuronal populations that are affected are not specified.

We agree with this limitation. The neuronal populations affected are those derived from apical progenitors in the neocortical VZ at the time of electroporation and their progeny. This includes postmitotic neurons and basal progenitors. However, the number of Tbr2 positive basal progenitors in the electroporated area was not affected by DN FGFRs (new Figure 2A). In addition, there was no change in the number of Satb2 positive neurons destined for the upper CP, or in *Sox2* positive apical neural stem cells. Cell proliferation and cell survival were also unaffected (new Figure 2A).

2) There is no direct evidence that the FGFR family members act redundantly during the transition from the multipolar to bipolar stage and migration into the cortical plate.

*FGFR1*‐3 are all expressed in the developing neocortex. Single mutants have subtle defects and the triple mutant has no telencephalon, precluding investigation of neuron migration (Paek et al., 2009; Iwata and Hevner, 2009). However, our use of dominant negatives, which inhibit the family by forming heterodimers (Ueno et al., 1992), suggests that one or more family members is important for migration. Moreover, each family member has the ability to rescue Rap1‐inhibited neurons. They thus act interchangeably under these conditions. The Wikipedia definition of genetic redundancy is “situations where a given biochemical function is redundantly encoded by two or more genes”. However, since we have not been able to formally prove that each family member functions in multipolar migration when expressed at endogenous level in vivo, we now say the evidence “suggests” that the FGFR family members act redundantly.

Can the dominant negative FGFR dimerize with receptors other than FGFR family members or N-Cad? Such interactions could contribute to the phenotypes observed.

The dominant‐negative mutants that were used have an intact extracellular domain but truncated cytoplasmic domain. They have the potential to interfere with endogenous FGFRs by competing for extracellular partners, including FGFs, NCad, NCAM, L1CAM and NEGR1, or by inhibiting endogenous FGFR kinase activation by forming mixed heterodimers. However, ability of both NCad and FGFR to rescue migration of Rap1 inhibited neurons, coupled with the lack of rescue by NCadΔEC4 and the finding that dominant‐negative NCad inhibits Reelin‐induced FGFR activation in neurons makes us favor NCad as the relevant partner.

3) Does NCad-dEC4 bind to receptors other than FGFRs?

We are not aware of any studies. However, we found that NCadΔEC4 still formed homophilic dimers with NCad, so it is not grossly misfolded.

4) Assays to identify binding partners, e.g. IP/mass spec experiments using lysates from primary cortical neurons expressing the dominant negative FGFR and NCad-dEC4 proteins, would be more informative.

In a different set of experiments, we did undertake large scale mass spec screens for neuronal proteins that bind NCad^W161A^ (the mutant that does not form *trans* homodimers) but not ECad. We used both BioID (with extracellular BirA) and affinity purification mass spec. Neither screen detected additional cell surface receptors.

5) While the HEK293T cells are a great starting point for the biochemistry, it is critical to perform the protein interaction and expression assays using purified cortical neurons or at least cultured primary cortical neurons. The fact that these changes, or lack of changes in some cases, occur in HEK293T cells does not guarantee that this is the case in post-mitotic cortical neurons.

The interaction, ubiquitination and protein degradation experiments were indeed performed in HEK293T cells because this kind of approach is very difficult in cultured neurons. HEK293T cells share features with neurons and are easy to transfect (Shaw et al., 2002). Nevertheless, we have added new data to the paper showing that endogenous *FGFR1* protein level increases when NCad is overexpressed in cultured cortical neurons (new Figure 3C), that inhibiting Rap1 in neurons in utero decreases FGFR protein level (new Figure 3B) and that the effect of Reelin on FGFRs in cortical neurons depends on NCad and Rap1 (new Figure 8D).

6) There appears to be a disconnect between the Rap1/CDH2/FGFR and Reelin/FGFR/ERK1 and 2 signaling pathways in this manuscript that are bridged by the following sentence and the authors' previous paper "The Rap1-dependent upregulation of NCad is triggered by Reelin, an extracellular ligand present in the MMZ (Jossin and Cooper, 2011)". This needs to be more clearly stated.

We agree that the link was unclear. We have performed a new experiment which demonstrates that NCad is important for the effect of Reelin on FGFR protein level and Erk phosphorylation (Figure 8D).

7) Why is there no control image for Figure 1B to show the Golgi orientation of control cells?

A control image has been added.

8) Figure 2A and 3D: Why is FGFR expression not visible in all lanes on the WB when the protein is overexpressed? If it is a matter of exposure time, it would be preferable to use longer exposures in order to see bands in all lanes that indicate FGFR expression. It would also be helpful to see loading controls like in Figure 3D.

The requested long exposure for FGFR has been added along with the loading control in new Figure 3—figure supplement 1. Long exposure for FGFR has been added to the new Figure 8A and B.

9) Figure 3A and 3B: In contrast to Figure 2C, there is no increase in FGFR1-Myc when co-expressed with NCad-HA. If the DNA concentration is adjusted as noted in the legend, it would still be expected that the NCad-HA and NCad-W161A-HA lanes show similar levels of FGFR1-Myc, which they do not. FGFR1 appears even lower when co-expressed with NCad-HA. On the other hand, NCad-dEC4, which is proposed to not interact with FGFR1, seems to increase FGFR1 expression. I would suggest to show blots with consistent data, or remove those claims in the text.

We varied the amount of FGFR DNA to compensate for FGFR stabilization by different cadherins and make sure we had enough protein to test for co‐immunoprecipitation. However, as the reviewer noted, we over‐compensated in the experiments shown. We have repeated the experiments and now show data where FGFR is equally expressed (Figure 5A, Figure 6—figure supplement 1).

10) The authors state that NCad-dEC4 does not interact with FGFR1 and is unable to rescue migration defects in Rap1GAP cells, which indicates that NCad interacts with FGFR1 through its EC4 domain. However, the data in Figure 5 shows that NCad increases FGFR1 expression and promotes neuronal migration through its EC1-2 domains. The same figure also shows that the EC4 domain-containing chimeric E/NCad does not increase FGFR or promote migration. This is confusing. If NCad does interact with FGFR1 via EC4 and promotes migration via EC1-2, it would suggest that homophilic binding of NCad or heterophilic binding to other proteins, in addition to heterophilic NCad:FGFR binding, plays a role in migration. Nevertheless, the authors present data indicating that homophilic NCad interactions do not have a role migration. These data need clarification.

This is exactly what we are claiming in this manuscript. Binding of NCad EC4 to FGFR is necessary but not sufficient to inhibit FGFR degradation. Indeed, ECad also interacts with FGFR through its EC4 but is unable to prevent FGFR degradation or promote neuronal migration in vivo. An NCad molecule containing EC4‐5 from ECad still protects FGFR from degradation and promotes neuron migration in vivo. The reciprocal switch, putting EC4 of NCad into ECad, does not give to ECad the ability to prevent FGFR degradation or to promote neuronal migration in vivo. The protection from degradation and promotion of migration map to EC1‐2 even though it needs the NCad‐FGFR interaction through EC4. This EC1‐2 function is not NCad‐NCad homophilic binding. We suggest that another protein interacting with NCad but not ECad EC1‐2 might be involved. Discovering this third partner is however beyond the scope of this manuscript.

11) Were the ERK1/2 constructs expressed in cortical neurons selectively expressed in post-mitotic cells?

Yes, we used the NeuroD promoter, as stated in the figure legend (Figure 9A).

12) The Discussion ends rather abruptly.

We thank the reviewer for the comment, and have added a concluding paragraph.

[Editors' note: the author responses to the re-review follow.]

Kon et al. revised their manuscript ʺFGFR ubiquitination and degradation controls neuronal migration in vivoʺ to reflect their observation that cortical neurons appear to stall at the multipolar stage when FGFR signaling is perturbed by transgenic expression of DN FGFR. They have done a nice job of streamlining the biochemical and in utero electroporation epistasis experiments, and performing more of the biochemical experiments in cultured cortical neurons as requested, to work out the signaling mechanism involved. While the manuscript overall has improved, there are several issues that I feel do not make it suitable for publication in eLife at this time and may be more suitable for a more specialized journal.First, I still worry that expression of a DN FGFR, as opposed to conditional genetic loss of function experiments, may have some off-target effects. In addition, the finding by Szczurkowska et al., 2018, that FGFR2 regulates neuronal migration and spine density suggests that the FGFRs may not act redundantly.

We addressed the issue of specificity of FGFRs downregulation and overlapping functions of FGFRs in multipolar migration. We specifically knock‐down *FGFR1*, 2 and 3. We tested at least 6 shRNA for each receptor and selected the most efficient. The efficiency of the shRNAs was tested in cell culture and in vivo as explained in Figure 1—figure supplement 1.

We found that the in vivo knock‐down of *FGFR1* and 2 induces an arrest of cells at the MMZ, with a more pronounced phenotype when the two receptors are downregulated together (Figure 1B). The knock‐down of FGFR3 resulted in a small, statistically non‐significant effect on cell positioning (Figure 1B). These results suggest that FGFRs work redundantly with a prominent role for *FGFR1* and *FGFR2*.

Reviewer #2 was also concerned that FGFR(DN) could affect NCad. We found that in cell culture, FGFR(DN) does not change NCad protein expression level, does not reduce NCad homophilic interaction and does not prevent NCad accumulation at cell‐cell junctions (Figure 3—figure supplement 1B).

Second, I do not agree that the terminology "multipolar migration" is justified without more detailed analysis of the cellular mechanism involved. Better high magnification/resolution images of cell morphology at this stage, combined with live imaging, would be required to determine why and how these cells are stalling.

To gain insight into the mechanism underlying the migration defect, we analyzed the morphology of migrating neurons. Analysis of the morphology revealed no difference in the number of neurites or in the cell body length‐to‐width ratio of FGFR‐inhibited multipolar neurons compared to control multipolar neurons

(Figure 2B-D). In addition, the few FGFR‐inhibited bipolar neurons migrating in the RMZ exhibited no difference in the length of the leading process and the cell body length‐to‐width ratio compared to control cells and possess an axon at the rear (Figure 2G=I).